


**Simple rules to minimize exposure to coseismic landslide hazard**
David G. Milledge[1], Alexander L. Densmore[2], Dino Bellugi[3], Nick J. Rosser[2], Jack Watt[2], Gen Li[4],
Katie J. Oven[2]
1. School of Engineering, Newcastle University, Newcastle upon Tyne, UK
2. Institute of Hazard, Risk, and Resilience and Department of Geography, Durham University,
Durham, UK
3. Department of Geography, University of California, Berkeley, USA
4. Department of Earth Sciences, University of Southern California, Los Angeles, USA
**Abstract**
Landslides constitute a hazard to life and infrastructure, and their risk is mitigated primarily by
reducing exposure. This requires information on landslide hazard at a scale that can enable informed
decisions about how to respond to that hazard. Such information is often unavailable to, or not easily
interpreted by, those who might need it most (e.g., householders, local government, and NGOs). To
address this shortcoming, we develop simple rules to identify landslide hazard that are
understandable, communicable, and memorable, and that require no prior knowledge, skills, or
equipment to evaluate. We examine rules based on two common metrics of landslide hazard, local
slope and upslope contributing area as a proxy for hillslope location, and we introduce and test two
new metrics: the maximum angle to the skyline and the hazard area, defined as the upslope area
with slope >39˚ that reaches a location without passing over a slope of <10˚. We then test the skill
with which each metric can identify landslide hazard - the probability of being hit by a landslide -
using inventories of landslides triggered by six recent earthquakes. We find that the maximum skyline
angle and hazard area provide the most skilful predictions, and these results form the basis for two
simple rules: 'minimize your maximum angle to the skyline' and 'avoid steep (>10˚) channels with
many steep (>39˚) areas that are upslope'. Because local slope alone is a skilful predictor of landslide
hazard, we can formulate a third rule as 'minimise local slope, especially on steep slopes and even
at the expense of increasing upslope contributing area, but not at the expense of increasing skyline
angle or hazard area'. Upslope contributing area, by contrast, has a weaker and more complex
relationship to hazard than the other predictors. Our simple rules complement, but do not replace,



detailed site-specific investigation; they can be used for initial estimation of landslide hazard or guide
decision-making in the absence of any other information.
**Keywords:** coseismic, landslide, heuristic, hazard, exposure
**1.  Introduction**
Landslides involve the downward movement of soil or rock under gravity, sometimes mixing with
water or air to run out rapidly over long distances. Landslides have considerable destructive potential
and constitute a major hazard to life and infrastructure (e.g. Alexander 2005; Petley, 2012; Klose et
al., 2016; Mertens et al., 2016).
Landslide risk can be mitigated by either reducing exposure - the likelihood that a particular person
or structure is hit by a landslide - or by reducing the consequences of landslide impact. The latter is
expensive for a building (Fell et al. 2005; Volkwein et al., 2011; Guillard-Gonçalves et al., 2016) and
extremely difficult for a person (Petley, 2012, Kennedy et al., 2015). As a result, efforts in reducing
landslide risk tend to focus on reducing exposure, primarily by siting infrastructure and assets (or
choosing to spend time) in places of lower landslide hazard. These choices, however, require
information on landslide hazard at a scale that can enable informed decisions about how to respond
to that hazard.
Quantitative landslide hazard information is commonly expressed as a relative weighting or
probability of landslide occurrence in a given location and over a specified period of time. This is
often communicated as a hazard map (Dransch et al., 2010). These maps can provide useful
information to inform decisions such as siting infrastructure, allocating resources, designing
countermeasures, or planning mitigation measures such as evacuation routes. There are, however,
at least five limitations to reliance on hazard maps as the sole source of landslide hazard information.
First, landslide hazard maps do not exist for all hazardous locations since their generation requires
technical expertise and site-specific information that may not be available. Second, where maps do
exist they may not be available to those that need them. Whether in physical or digital form, hazard
maps are rarely held by the communities that live within their boundaries (Alexander, 2005; Mills and
Curtis, 2008; Twigg et al., 2017). Third, where landslide hazard maps are available their resolution

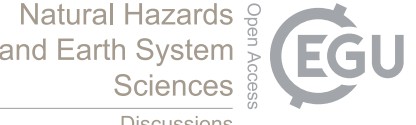



may not be fine enough to address the questions that potential users will have. In everyday decisions,
from where to build a house to which way to walk, metres matter for determining landslide exposure.
Landslide hazard varies over very short length scales (tens of metres), but national- or even regional-
scale hazard maps cannot resolve hazard at those scales, and hazard maps at the appropriate scale
would be extremely costly and time-consuming to produce over large areas. Fourth, landslide hazard
maps are designed for technical users (such as engineers and planners) and thus can be difficult for
non-technical users to interpret (Dransch et al., 2010). Hazard is often expressed in probabilistic
terms, which are inherently difficult to communicate and understand (Thompson et al., 2015). The
maps may also require particular equipment, such as a computer with appropriate software, or
additional contextual information to enable clear visualisation or orient the user (Mills and Curtis,
2008). Finally, landslide hazard maps may lack appropriate information for decision-making. For
example, landslide hazard is commonly equated simply with the probability of landslide initiation at
a given location, rather than the probability that that location is impacted by a landslide occurring
there or somewhere upslope.
In the absence of detailed hazard maps, how should we make decisions about siting infrastructure
or spending time in landslide-prone areas? An alternative form of hazard information might be a set
of general rules that can be memorised by anyone who might be exposed to landslide hazard, or by
those charged with managing landslide risk, to be applied where no other information exists. A good
general rule should: 1) be understandable, communicable and memorable; 2) require no prior
knowledge, skills or equipment to evaluate; 3) be a skilful discriminant of hazard; and 4) be cast so
that it does not increase exposure to another hazard. A good example of such a rule would be the
instruction to minimise exposure to tsunami: "in case of earthquake, go to high ground or inland"
(Atwater et al., 1999, p20). Research has shown that these types of simple rules are already to some
extent implicitly coded into the decisions that people make (e.g. Gigerenzer, 2008), reflecting tacit
knowledge of hazards (e.g. Shaw et al., 2008; Lebel, 2013; Twigg et al., 2017). Importantly, however,
there are limits to this tacit knowledge (Briggs, 2005); in particular, the body of experience required
to generate these rules is limited by both the infrequency of triggering events, such as earthquakes
or large storms, and a focus on *normal* rather than *unusual* but not improbable events, introducing
biases (McCammon, 2004; Kahneman and Klein, 2009). For example, while perennial rainfall-





triggered landslides and the risks that they pose may be familiar to people in landslide-prone
communities, landslides triggered by large earthquakes may fall outside of residents' lived
experience, and so will be more challenging to comprehend and account for in decision-making. If
simple, memorable rules (fulfilling criteria one and two) could be derived from a large inventory of
hazardous events, these biases might be reduced while maintaining the other benefits of a rule-
based approach (criteria three and four). Such a set of data-based rules could be used in the
absence of, or in conjunction with, existing tools such as hazard maps and local knowledge, both to
inform decisions and to inspire discussion amongst householders, local government, and non-
governmental organisations. Such knowledge is commonly in demand not only from technical users
but from lay people (Twigg et al., 2017; Datta et al., 2018), especially because self-recovery after
disasters is increasingly recognised as a critical mechanism of recovery (Twigg et al., 2017).
Here we focus on rules that can be derived from the topography surrounding a given location and
that differentiate exposure to coseismic landslide hazard on a scale of tens to hundreds of metres.
Such rules are likely to be most useful for decisions before an earthquake about where to site
infrastructure or spend time, and may be of less use for decisions during an earthquake when time
is limited. We focus on earthquakes because landsliding is an important, but poorly understood,
aspect of hazard in many recent continental earthquakes (Huang and Fan, 2013; Roback et al.,
2017). Some of our results may be transferrable to landslides caused by more frequent triggers,
such as storms, and we consider this point in the discussion.
We examine candidate rules based on our existing understanding of landslide mechanics to identify
those that meet criteria one and two above. We then test the skill with which each candidate rule
can identify landslide hazard using inventories of coseismic landslides from the recent Finisterre,
Northridge, Chichi, Wenchuan, Haiti, and Gorkha earthquakes. Our goal is to determine the rule or
rules that best fulfil the four criteria listed above, and that therefore provide the best combination of
simplicity and skill in anticipating coseismic landslide impacts. We ask two key questions: (1) to what
extent could observed landslide locations have been predicted by these simple rules alone, without
recourse to more complex models; and (2) is there a single rule or set of rules that performs well
across all earthquakes, and could form the basis for anticipating landslide-affected locations in a
future earthquake? While patterns of landsliding in these earthquakes have been previously





established, this is to our knowledge the first attempt to extract a more general set of rules from the

combined datasets.

This paper is necessarily technical, addressing the question of whether it is possible to formulate

such rules, identifying which rules work best and assessing their performance. We therefore expect

the paper's primary audience to be technical experts with an interest in landslide risk reduction. We

have begun to explore ways of expressing these rules in a format that is more accessible to a general

audience (e.g. Milledge et al., 2018).

## 2. Potential predictors for coseismic landslide hazard: slope and upslope contributing area

Local slope has been identified as an important driver of landslide occurrence in almost all landslide

studies (e.g. Harp et al., 1981; Tibaldi et al., 1995; Keefer, 2000; Wang et al., 2003; Xu et al., 2012,

2013; Parker et al., 2017). This is consistent with mechanistic expectations based on the balance of

driving and resisting forces on an inclined failure plane (Taylor, 1937). Local slope is an intuitive

parameter that is familiar to most people and can be easily estimated in relative terms (i.e., hillside

A is steeper than hillside B) without specialised equipment. Shaking intensity is commonly identified

as the other dominant control on coseismic landslide occurrence. However, shaking for any future

earthquake cannot be predicted due to lack of certainty on source location, magnitude, rupture style,

and local site effects. It is therefore difficult to incorporate into a general rule for future landslide

hazard.

Ridges are often considered to be areas of high coseismic landslide probability due to topographic

amplification (Densmore and Hovius, 2000), while rivers are by definition areas of flow concentration

into which landslides from multiple potential initiation zones may run out. Here we use upslope

contributing area as a continuous estimator of the proximity to a ridgeline (defined here as an area

with no upslope cells) or a valley, in order to assess how hazard may vary with position in the

landscape.

Other predictors have been identified in coseismic landslide studies, but these generally have a

secondary effect and are not consistently identified as controls (Parker et al., 2017). Elevation and

aspect in particular lack a consistent explanation or pattern as a control on coseismic landslide





hazard (Parker et al., 2017). Other common predictors are difficult to evaluate 'on the ground' without
specialised equipment or knowledge. Soil type, rock type, or land cover may be relevant to slope
stability but are difficult to identify without specialised training. Curvature is strongly dependent on
the length scale over which it is measured and is extremely difficult to estimate by eye, particularly
in rough natural topography. Proximity to roads is often possible to estimate in the field, but inclusion
of this factor assumes that all roads are similar in their design, age and construction, and thus have
similar impacts on slope stability.

**3. Accounting for runout in landslide hazard: reach angle and runout routing**
All of the potential predictors described above are linked to the probability of coseismic landslide
initiation. Once triggered, however, landslide material may run out for long distances and over large
areas. Thus, there are substantial portions of any landscape where landslide initiation is unlikely but
where contact with a landslide is still possible – for example, at the foot of a steep hillslope.
Mechanistic modelling of landslide runout is computationally intensive and strongly sensitive to initial
conditions, taking it beyond the capacity of exposed communities (e.g., George and Iverson, 2014).
In contrast, simple empirical approaches that have shown some predictive power fall into two
categories: reach angles and runout routing.
The Fahrboschung or reach angle from the crown of the landslide to the toe of its deposit has been
shown to follow an exponential decrease with landslide volume (Heim, 1882; Corominas, 1996;
Hunter and Fell, 2003). The reach angle concept has been incorporated into a small number of
hazard maps as a way to represent the probability that a landslide will reach a given location, and
can be coupled with predictions of the probability of landslide initiation (e.g. Kritikos et al., 2015).
However, these complex combinations of probability are difficult to distil into a single simple rule and
to our knowledge, this has not yet been done.
If initiation probability is unknown and we make the conservative assumption that any cell can initiate
a landslide, then the hazard at a given location becomes proportional to the area that protrudes
above a cone with its apex at the location of interest and its sides inclined at a critical reach angle
from the horizontal. This approach has similarities with local sloping base level (Jaboyedoff et al.,
2004) and excess topography metrics (Blöthe et al., 2015), which both project surfaces through the





landscape to identify less stable zones, though neither of these approaches are framed in terms of
reach angles. Even this simple approach, which neglects initiation probability, is hard to distil: 1) its
conceptual complexity makes it difficult to communicate; 2) its predictions depend on a reach angle
parameter that is poorly constrained; and 3) the area protruding from an imaginary surface projected
beneath the land surface is very difficult to estimate by eye, particularly where significant areas may
be occluded from the viewpoint. An alternative metric would simply be the maximum angle from the
horizontal to the skyline, which can be interpreted as the maximum (or worst-case) reach angle for
that location. This metric is much simpler and thus easier to communicate and remember, can be
estimated by eye, and avoids the problem of choosing a critical reach angle.
Runout routing approaches assess the probability that landslide debris will reach a given location by
assuming that it flows downslope and that its probability of stopping is dependent on some local
property of the path along which it flows. This approach ranges in complexity from detailed physics-
based treatments (George and Iverson, 2014; von Ruette et al., 2016) to simple empirical rules such
as the local slope or junction angle of flowpaths (Benda and Cundy, 1990; Fannin and Wise, 2001;
Montgomery and Dietrich, 1994; Densmore et al., 1998). Hazard estimates are then a function of the
initiation probability integrated over the upslope area and the stopping probability for each potential
event. To incorporate these considerations as simply as possible, we introduce a new approach
(described below) that accounts for local slope at both the locations of landslide initiation and along
the flow path. While this approach does not capture the dynamic behaviour of landslide initiation or
runout, we include it so that we can test the skill of such non-local approaches and the need to
account for them in our simple rules.

**4. Earthquake inventories**
**4.1.     1994 $M_w$ 6.7 Northridge**
Topographic relief and seismicity in southern California are associated with dextral transpression at
the Pacific-North America plate boundary (Montgomery, 1993). The study area lies within the
western Transverse Ranges of southern California and is largely underlain by weakly cemented
sedimentary rocks except for the mainly granitic and gneissic San Gabriel and Verdugo mountains
and stronger sedimentary rocks in the Simi Hills (Colburn et al., 1981; Tsutsumi and Yeats, 1999;



Parise and Jibson, 2000). Estimated denudation rates for the Santa Monica and San Gabriel
mountains are 0.1-1 mm/yr (Meigs et al., 1999; Lave and Burbank, 2004). The region has a warm-
summer Mediterranean climate (Peel et al., 2007) with monthly average temperatures ranging from
1 - 18 ˚C (NOAA, 2017) and mean annual precipitation of 0.3–0.9 m (National Atlas of United States,
2011). Vegetation is predominantly annual grassland, sage scrub, and chaparral with some piñon-
juniper, oak and pine woodlands (Griffith et al., 2016).
The $M_w$ 6.7 Northridge earthquake occurred on 17 January 1994 and ruptured 14 km of a south
dipping (35°) blind thrust fault with a hypocenter at 19 km depth (Wald and Heaton, 1994, Hauksson
et al., 1995). The earthquake produced recorded ground accelerations of up to 2 $g$ (Harp and Jibson,
1996) and maximum surface displacements of ~4 m. More than 11,000 landslides were triggered
across a total area of ~10,000 km$^2$ (Harp and Jibson, 1996). Landslides were mapped immediately
after the earthquake using field studies and aerial reconnaissance and were manually digitized on
1:24,000 scale base maps. Landslides >10 m across could be confidently identified and location
errors were estimated to be <30 m (Harp and Jibson, 1996).

**4.2.    1993 $M_w$ 6.9 Finisterre**
Oblique convergence of the Australian and Pacific plates has driven uplift of the Finisterre Mountains
to an elevation of ~4 km since 3.7 Ma (Abbott et al., 1997). The Finisterre Mountains consist of
volcanic and volcaniclastic rocks thrust over coarse-grained foreland deposits and capped by
limestones (Davies et al., 1987; Abbott et al., 1994). Denudation rates in these mountains are up to
0.3 mm/yr averaged over the time of range formation (Abbott et al., 1997). The region has a tropical
climate (Peel et al., 2007), with high and stable monthly average temperatures (26-27˚C) and mean
annual precipitation ranging from ~2.5 m in the west to ~4 m in the east (Hovius et al., 1998). The
vegetation is predominantly tropical wet or tropical montane evergreen forest with sub-alpine
grasslands on some of the higher peaks (MacKinnon 1997; Paijmans 1975).
A $M_w$ 6.9 earthquake occurred on 13 October 1993, with a hypocentre at 25 km depth, rupturing the
north-dipping Ramu-Markham thrust fault to within a few hundred meters of the surface (Stevens et
al., 1998). The event was followed by multiple aftershocks (5 > $M_w$ 6) including a $M_w$ 6.7 event on 25
October 1993 with a hypocentre at a depth of 30 km. About 4,700 landslides with a total surface area



of about 55 km$^2$ were triggered by these earthquakes and were mapped from 30 m resolution SPOT
images (Meunier et al., 2007).

### 236     4.3.     1999 M$_w$ 7.6 Chi-Chi

Taiwan's mountains are the product of oblique collision between the Philippine Sea plate and the
Eurasian continental margin. The study area lies within the central mountains of Taiwan and is
largely underlain by Neogene sediments and older metasedimentary rocks (Lin et al., 2000).
Denudation rates in the central mountains of Taiwan are high, averaging 3-7 mm/yr (Dadson et al.,
2003). The region has a humid subtropical climate (Peel et al., 2007) with a mean annual
temperature of 22˚C, a mean annual precipitation of 2.5 m and an average of four typhoons per year
(Wu and Kuo, 1999). Subtropical moist broadleaf forests occupy most of the island including its
mountainous interior (Olsen et al., 2001).
The M$_w$ 7.6 Chi-Chi earthquake occurred on 21 September 1999 with a hypocentre at 8–10 km
depth, rupturing ~100 km of the east-dipping Chelungpu thrust fault (Shin and Teng, 2001). The
earthquake produced recorded ground accelerations of up to 1 $g$ (Lee et al., 2001) and maximum
surface displacements of ~8 m (Chi et al., 2001; Shin and Teng, 2001). The earthquake triggered
more than 20,000 landslides with the majority occurring across a 3,000 km$^2$ region (Dadson et al.,
2004). Landslides were mapped by the Taiwan National Science and Technology Centre for Disaster
Prevention from SPOT satellite images with a resolution of 20 m; landslides with areas >3,600 m$^2$
were resolved, with location errors estimated to be ~20 m (Dadson et al., 2004).

### 254     4.4.     2008 M$_w$ 7.9 Wenchuan

The Longmen Shan mountain range defines the eastern margin of the Tibetan Plateau with
displacement taken up mainly on oblique dextral-thrust faults (Burchfiel et al., 1995; Densmore et
al., 2007). The Longmen Shan are underlain by a complex lithological assemblage comprising
Proterozoic granitic massifs, a Palaeozoic passive margin sequence, a thick Triassic-Eocene
foreland basin succession, and minor exposures of poorly-consolidated Cenozoic sediment
(Burchfiel et al., 1995). Denudation rates are estimated at ~0.5 mm/yr over decadal to millennial





timescales (Ouimet et al., 2009; Godard et al., 2010; Liu-Zeng et al., 2011). The region has a humid
subtropical climate (Peel et al., 2007), with an annual average temperature of 15-17 °C and average
annual rainfall varying from ~1100 mm at the margin to ~600 mm on the plateau, of which 70%–80%
falls from June to September (Liu-Zeng et al., 2011; Li et al., 2016). The natural vegetation is
montane broad-leaved and conifer forest below 4000 m with alpine shrub land and steppe vegetation
at higher elevations (Yu et al., 2001).
The $M_w$ 7.9 Wenchuan earthquake occurred on 12 May 2008, rupturing ~320 km of the steeply
northwest-dipping Yingxiu-Beichuan and Pengguan faults (Xu et al., 2009). It had an oblique dextral-
thrust focal mechanism with a hypocentre at 14-19 km depth. The earthquake produced recorded
ground accelerations of up to 1 *g* (Li et al., 2008) and maximum vertical and dextral displacements
of 6.2 m and 4.5 m, respectively (Liu-Zeng et al., 2009; Gorum et al., 2011). The earthquake triggered
more than 60,000 landslides across a total area of 35,000 km$^2$ (Gorum et al., 2011; Li et al., 2014).
We used a subset of the landslide inventory compiled by Li et al. (2014), who mapped landslides
from high-resolution (<15 m) satellite images and air photos. The subset of 18,700 landslides (all
mapped landslides east of 104 E), was chosen to avoid gaps in the 30 m resolution SRTM
topographic data. Location accuracy for landslides is thought to be similar to the pixel size of the
satellite images used, ~15 m (Li et al., 2014).

**4.5.        2010 $M_w$ 7.0 Haiti**
Haiti's mountains are the product of oblique convergence between the Caribbean and North
American plates (Pubellier et al., 2000). The study area is underlain by northwest-southeast oriented
sub-parallel belts of igneous, metamorphic and sedimentary rocks (Sen et al., 1988, Escuder-Viruete
et al., 2007). Mean elevation and relief generally increase from north to south, to a plateau at ~2500
m (Gorum et al., 2013). The region has a tropical climate (Peel et al., 2007) with a mean annual
temperature of 25°C and   mean annual precipitation of ~1.2 m, with two rainy seasons per year
(April-June and October-November) and hurricanes between June and November (Gorum et al.,
2013; Libohova et al., 2017). The study area lies predominantly within the moist broadleaf forest





biome with some pine or dry broadleaf forest (Olsen et al., 2001) but also has extensive (~50%
by area) savannah, shrub or herbaceous cover (Churches et al., 2014).
The $M_w$ 7.0 Haiti earthquake occurred on 12 January 2010, with a hypocentre at 13 km depth but
without any detectable surface rupture (Mercier de Lépinay et al., 2011). The complex rupture
involved both the Léogâne blind thrust fault, responsible for ~80% of the seismic moment (Hayes et
al., 2010) as well as deep lateral slip on the Enriquillo–Plantain Garden Fault (Hayes et al., 2010,
Mercier de Lépinay et al., 2011). The earthquake triggered more than 30,000 landslides across a
3,000 km² region (Xu et al., 2014). We used an inventory of 23,679 landslides mapped by Harp et
al. (2016) from publicly available satellite imagery with a resolution 0.6 m before and after the
earthquake; landslides with areas >10 m² were resolved (Harp et al., 2017).

**4.6.     2015 $M_w$ 7.8 Gorkha**
The Himalayas are the product of active continental convergence of India and Asia, much of which
is accommodated by the seismogenic Main Himalayan Thrust (Lavé and Avouac, 2000). The study
area is underlain by variably metamorphosed sedimentary and igneous rocks of Proterozoic and
early Paleozoic age with Paleozoic and Mesozoic sedimentary rocks and low-grade
metasedimentary rocks to the north marking the southern margin of the Tibetan Plateau (Hodges et
al., 1996; Searle and Godin, 2003; Craddock et al., 2007). Denudation rates in the study area range
from 0.3-3 mm/yr over millennial time scales (Lupker et al., 2012; Godard et al., 2014). Mean annual
temperature varies with elevation across the study area from ~18˚C in the valley bottoms to -6˚C at
high elevations. Average annual rainfall is also topographically controlled, ranging from ~1 m/yr at
the range front to >3 m/yr in two bands along the southern margins of the Lesser and Greater
Himalaya to <0.5 m/yr on the Tibetan plateau (Bookhagen and Burbank, 2006). Natural vegetation
is dominated by temperate broadleaf and coniferous forests up to 3000 m with alpine tundra above
the tree line (Singh and Singh, 1987).
The $M_w$ 7.8 Gorkha earthquake occurred on 25 April 2015, rupturing ~140 km of the north-dipping
Main Himalayan Thrust (Hayes et al., 2015; Elliott et al., 2016). It had a hypocentre at 8.2 km depth
but did not rupture to the surface (Hayes et al., 2015). The event was followed by a series of large





aftershocks, including a $M_w$ 7.2 event on 12 May which ruptured a portion of the Main Himalayan
Thrust directly east of the 25 April rupture (Avouac et al., 2015). The earthquake triggered
approximately 25,000 landslides with a total surface area of about 87 $km^2$ (Roback et al., 2017). We
used an inventory of 24,915 landslides mapped by Roback et al. (2017) from Worldview-2
Worldview-3 and Pleiades imagery, with a resolution <0.5 m, before and after the earthquake.

5. **Methods**
**5.1.        Conditional probability**
Landslide hazard can be defined as the probability of being hit by a landslide in a given location
and within a given time window (Lee and Jones, 2004). Here we make no distinction between
consequences of being hit by landslides of different sizes or velocities, assuming that all are
equally dangerous. This probability can be expressed mathematically as P(L|x,y,t), where L is the
outcome of being hit by a landslide, *x,y* are the coordinates for a particular location and t is the time
window of interest. We do not address the timing of landsliding, assuming that this is driven by the
timing of an earthquake and is thus unpredictable (Geller, 1997). Instead we focus on landslide
susceptibility given an earthquake that produces shaking of unknown intensity at a location (*x,y*),
hence the notation P(*L|x,y*). We assume that the hazard at that location can be approximated by
some location-specific characteristic (*a*). Thus, the landslide hazard at (*x,y*) is the conditional
probability of being touched by a landslide given the value of the characteristic at that location,
P(*L|a*), and can be calculated using Bayes Theorem:

$$P(L|a) = \frac{P(L)\,P(a|L)}{P(a)} \qquad\qquad\qquad (1)$$

where *a* is a specific characteristic of the location (e.g., the topographic slope). If we assume that
the relationships between past landslides and local characteristics are good predictors of their future
relationships then we can construct empirical conditional probability calculations from landslide
inventories. If we grid the topography, then the Bayes equation can be easily rewritten in terms of





the numbers of grid cells, and in this form the direct equivalence of landslide conditional probability
and landslide area density (e.g., Meunier et al., 2007; Dai et al., 2011; Gorum et al., 2014) is clear:

$$P(L|a) = \frac{N(a \cap L)}{N(a)}$$   (2)

where N($a \cap L$) is the number of cells with a given value of characteristic *a* that are touched by a
mapped landslide, N(*a*) is the number of cells with the characteristic of *a* in the entire study area,
and the study area is defined by the smallest convex hull that contains all of the observed landslides.
To account for variability in the magnitude of shaking between the six study areas, we normalise the
conditional probability of being hit by a landslide P(*L*|*a*) by the study area average probability of
landsliding P(*L*) to generate a relative hazard. This can be shown to be directly equivalent to the
'frequency ratio' (e.g., Lee and Pradhan, 2007; Lee and Sambath, 2006; Yilmaz, 2009; Kritikos et
al., 2015):

$$\frac{P(L|a)}{P(L)} = \frac{N(a \cap L)/N(a)}{N(L)/N(S)} = \frac{N(a \cap L)}{N(a)}\frac{N(S)}{N(L)}$$   (3)

where N(*S*) is the total number of cells in the study area and N(*L*) is the number of cells touched by
landslides. Our normalised conditional probability is also directly equivalent to the 'probability ratio'
used by Lin et al. (2008) and Meunier et al. (2008) since, from Bayes Theorem:

$$\frac{P(L|a)}{P(L)} = \frac{P(L)\,P(a|L)}{P(a)P(L)} = \frac{P(a|L)}{P(a)}$$   (4)

We display the normalised conditional probability on a logarithmic scale for readability, resulting in a
probability metric that is strongly similar to the 'information value' metric used in some landslide
susceptibility analyses (e.g., Yin and Yan, 1988).
Conditional probability analysis is advantageous for its direct link to hazard and does not require us
to impose a functional form to the data. However, the results are partly dependent on bin size and





location for the predictor variable, and bins with few observations (i.e. N($a$)<<N($S$)) can result in noisy
data that are difficult to interpret. To aid interpretation in the presence of noise, we fit cubic polynomial
functions to one-dimensional conditional probability data and a logistic function to two-dimensional
data. To highlight the parts of the data where we have few observations and thus where our
confidence in the results is lower, in the one-dimensional case we include a single bulk PDF of the
predictor variable on the x-axis below the conditional probability curve, and we limit ourselves to
calculating probability only where there are more than 10 observations per bin in the two-dimensional
case. Whilst other statistical approaches could be used here (e.g. Pradhan, 2013), our intention is
not to find the statistical approach that provides the most powerful synthesis of the different variables,
but to test the effectiveness of the variables themselves at distinguishing hazard when applied in the
form of simple rules.

382        **5.2.        Receiver operating characteristic curves**

Any simple rule for identifying more or less hazardous locations in the landscape will produce a
relative measure of landslide probability. To evaluate this measure against a binary landslide map
or inventory (where every cell is classified as landslide or non-landslide), it must be converted into a
binary classification. A common approach to this problem is to construct a receiver operating
characteristic (ROC) curve (e.g., Frattini et al., 2010). This curve quantifies both the benefit of a
given classification in terms of successfully classified outcomes (landslide and non-landslide
locations correctly identified, true positives and true negatives respectively) and also the cost (non-
landslides identified as landslides, false positives; and vice versa, false negatives). The ROC curve
is constructed by thresholding a continuous variable (e.g., slope) and calculating the true positive
rate as the number of true positives normalised by all positive observations, and the false positive
rate as the number of false positives normalised by all negative observations. Evaluation of these
rates at different threshold values results in a curve, where the 1:1 line reflects the naïve (i.e. random)
case. The area under the curve (AUC) tends to 1 as the skill of the classifier improves towards
perfect classification and to 0.5 as the classifier worsens towards the naïve (random) case. We
calculate ROC curves for all of our chosen predictive approaches for each inventory.





### 5.3. Topographic analysis

All of the metrics tested here are defined using topographic data in the form of digital elevation models (DEMs). We use 30 m resolution DEM data at all sites: for Northridge they are derived from the down-sampled 10 m NED elevation data (https://lta.cr.usgs.gov/NED), while for all other sites we use 1-arc sec Shuttle Radar Topography Mission (STRM) elevation data (http://srtm.csi.cgiar.org/).

#### 5.3.1. Slope and upslope contributing area

We calculate local slope as the steepest path to a downslope neighbour from each cell (Travis et al., 1975) because calculating slope over larger (e.g. 3x3 cell) windows for a 30 m resolution DEM results in considerable underestimation (Claessens et al., 2005). We calculate upslope contributing area using a multiple flow direction algorithm (Quinn et al., 1991) having filled pits using a flood fill algorithm (Schwanghart and Kuhn, 2010). These topographic analyses are performed in Matlab using TopoToolbox v1.06 (Schwanghart and Kuhn, 2010).

#### 5.3.2. Skyline angle analysis

To capture the effect of both initiation and runout we define the skyline angle as the maximum angle from horizontal to the skyline for a given location. This is easily estimated by eye in the field and can be interpreted as the maximum (or worst-case) reach angle for that location. It is a runout-dominated metric in that it does not take into account the probability of initiation.

For each cell in a study area we estimate the skyline angle by calculating vertical angles between the target cell and every other cell within a 4.5 km radius. This radius is chosen to exceed the dominant channel spacing for the study area with widest spacing (Wenchuan) and thus to fully capture the local skyline. For the Wenchuan study area the characteristic hillslope length, estimated following the method of Roering et al. (2007), is ~500 m. Thus a conservative estimate on dominant channel spacing would be ~1 km. We choose larger window size because skyline angle estimates become asymptotically insensitive to window size, so that the only constraint is run time. MATLAB code for the routine is included in the supplemental information. This approach is physically limited in at least two ways (Figure 1a). First, it does not account for the dependence of runout on the size of the initial failure or how the failure volume may increase or decrease during runout (e.g.

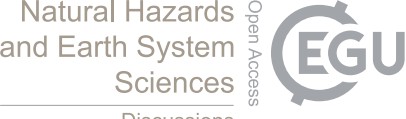

Corominas, 1996). Second, it does not honour flow paths. The skyline cell that generates the
steepest slope to the target cell does not have to be connected to the target cell by a flowpath with
monotonically decreasing elevation. However, this metric provides a measure of the gravitational
potential energy available to drive runout in the vicinity of the target cell.

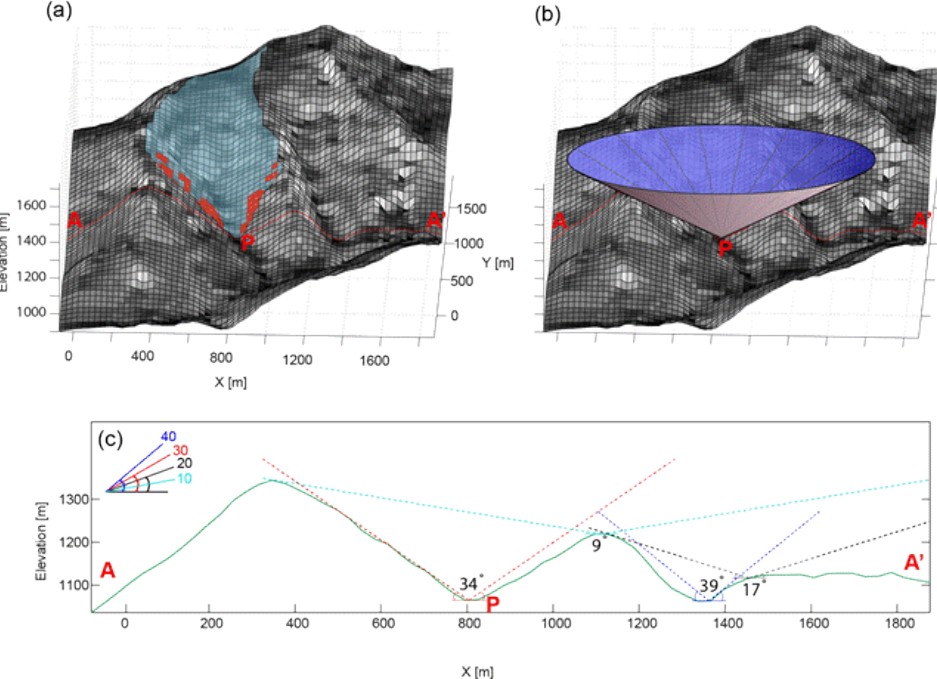


**Figure 1.** Schematic view of the different topographic metrics tested here. (a), perspective view of a
landscape with each cell shaded according to its local slope from light (steep) to dark (gentle). The
upslope contributing area for point P is coloured blue, and the cells steeper than 39˚ that have a flow
path to P that is never less than 10˚ are coloured red. (b), the same perspective view with a cone
projected from point A at an angle of 34˚ so that the surface of the cone is in places tangent to but
never intersects the ground surface, indicating a maximum skyline angle of 34˚ for point P. (c), cross
section A-A' through the landscape (highlighted in red on panels a and b) with dashed lines showing
skyline angles at four example locations.

### 5.3.3.  Runout routing analysis
To assess the importance of non-local runout paths on landslide probability, we follow the approach
of Dietrich and Sitar (1997) who proposed the simplest possible debris flow runout model, requiring





only thresholds to define instability and for downslope motion to continue. This simple model,
referred to as SHALRUN, was integrated with the coupled hydrologic-slope stability model
SHALSTAB in an efficient parallel framework to predict landslide hazard potential in California
(Bellugi et al, 2011). SHALRUN required only two field-calibrated parameters: a critical rainfall
threshold to define instability, and a minimum slope threshold for downslope motion to continue. To
apply this model in the context of coseismic landslides (SHALRUN-EQ) we modify the condition for
landslide initiation, replacing the critical rainfall threshold with a slope threshold. We thus assume
that landslide initiation and deposition are entirely dependent on the local slope of the ground surface
$\theta$ (i.e., landslides are more likely to initiate on steeper slopes and deposit on flatter slopes), further
increasing the simplicity of the model. More formally, SHALRUN-EQ predicts the upslope hazard
area $A_h$ as the upslope area weighted by the joint probability of landslide initiation and runout.
Locations with higher $A_h$ should have higher exposure to coseismic landslide hazard than those with
low (or no) $A_h$. Formulation of the model requires: (1) determination of the mobilisation probability at
each cell i in the study area ($P_{mi}$); (2) determination of the connection probability for mobilised
material from each cell i to the target cell j ($P_{cij}$); (3) convolution of (1) and (2) to get the locational
hazard ($P_{mcij}$); and (4) accumulation of the locational hazard to determine a hazard area above each
target cell j ($A_{hj}$).
In order to generate a simple rule, our model assumes that landslide initiation and deposition are
entirely dependent on the local slope of the ground surface $\theta$ (i.e. landslides are more likely to initiate
on steeper slopes and deposit on flatter slopes). For landslide initiation, we assume that slopes
above a threshold slope $\theta_m$ are all equally capable of initiating a landslide with probability $P_{mi}$:

$$P_{mi} = \begin{cases} 1 : & \theta_i \geq \theta_m \\ 0 : & \theta_i < \theta_m \end{cases} \qquad (5)$$


where $\theta_i$ is the observed local slope in a downslope direction at cell i and $\theta_m$ is the critical slope
required for landslide initiation.
In order to represent a landslide hazard, mobilised material must be able to runout from the initiation
point to the target cell j. This relationship is binary: either these points are connected by a viable

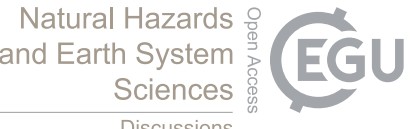

runout path or they are not. We assume that the flow path will follow the path of steepest descent.
This path must enable continued runout for its entire length; if at any point on the flow path the
material is fully deposited, then that initiation zone will be disconnected from cell j. Thus, the point
along a given flow path that is most likely to cause deposition becomes the controlling location for
the connection of all upslope points. Surface slope has been used to describe the probability that
landslide material entering a cell will be deposited rather than continuing into the next downslope
cell (e.g., Benda and Cundy, 1990; Fannin and Wise, 2001). For landslide deposition, we apply the
simplest possible stopping condition, and assume that landslide run-out ceases on slopes gentler
than a critical angle ($\theta_s$). The probability that a landslide initiated at point i reaches point j ($P_{cij}$) can
thus be expressed as:

$$Pc_{ij} = \begin{cases} 1: \theta min_{ij} \geq \theta_s \\ 0: \theta min_{ij} < \theta_s \end{cases} \qquad (6)$$


where $\theta min_{ij}$ is the minimum slope for the flow path from cell i to cell j, and $\theta_s$ is the critical slope
required for stopping.
We combine the initiation and runout probabilities to calculate the locational hazard $P_{mcij}$ as the area
($a_i$) in cell i weighted by the probability that a landslide is both mobilised in cell i and is connected to
cell j:

$$P_{mcij} = a_i \, P_{mi} \, P_{cij} \qquad (7)$$

Assuming that $\theta_s > 0$, we calculate the hazard area $A_{hj}$ for each target cell j by summing locational
hazard in the n cells upslope of j, normalised by the unit contour length to minimise grid resolution
bias:

$$A_{hj} = \sum_{i=1}^{n} \left( \frac{a_i}{l_j} P_{mi} \, P_{cij} \right) \qquad (8)$$





where $l_j$ is the unit contour length at $j$, calculated as $a_j^{0.5}$. Equation 8 is evaluated for every cell in the
study area to generate a spatial grid of hazard area $A_h$ (Figure 2). Our choice of step functions for
the mobilisation ($P_m$) and connection ($P_c$) probabilities allows us to interpret $A_h$ as the upslope area
per unit contour width with local slope steeper than $\theta_m$ from which a landslide will reach the cell of
interest by moving downslope along a path that is always steeper than $\theta_s$. Alternative formulations
could be used for $P_m$ and $P_c$ but these would result in a less intuitive index that would be difficult to
implement as a simple rule.

There is implicit resolution dependence to the stopping condition $\theta_s$ since it assumes that the low
gradient area is long enough (in terms of flow path length) that the landslide will stop. Similarly, there
is resolution dependence to the initiating condition $\theta_m$ as topographic surfaces will be more or less
smooth, depending on the resolution of the DEM (Classens et al., 2005). Also, the initiation
probability is based on local slope alone and so does not account for any of the other possible drivers
of coseismic landslide initiation, such as topographic amplification (Meunier et al., 2008), or pore
water pressure (e.g., Xu et al., 2012). While many more complex models exist that account for
initiation volumes and flow dynamics (e.g., George and Iverson, 2014; von Ruette et al., 2016), we
seek the simplest possible model that captures the effects of drainage networks in accumulating
hazard, of steep slopes in landslide initiation, and of gentle slopes in landslide deposition.
The model has two parameters ($\theta_m$ and $\theta_s$), both of which are effective rather than measurable. We
first optimise the model for each inventory to establish its performance under the best possible
scenario, where the model is fitted to the data. We then test the model using the average of the
optimised parameters from the six inventories to represent a more realistic application where these
parameters must be estimated from previous events. Thus, the values of $\theta_m$ and $\theta_s$ should not be
interpreted as mechanistic thresholds, but rather as the result of an optimization that also depends
on the DEM resolution.

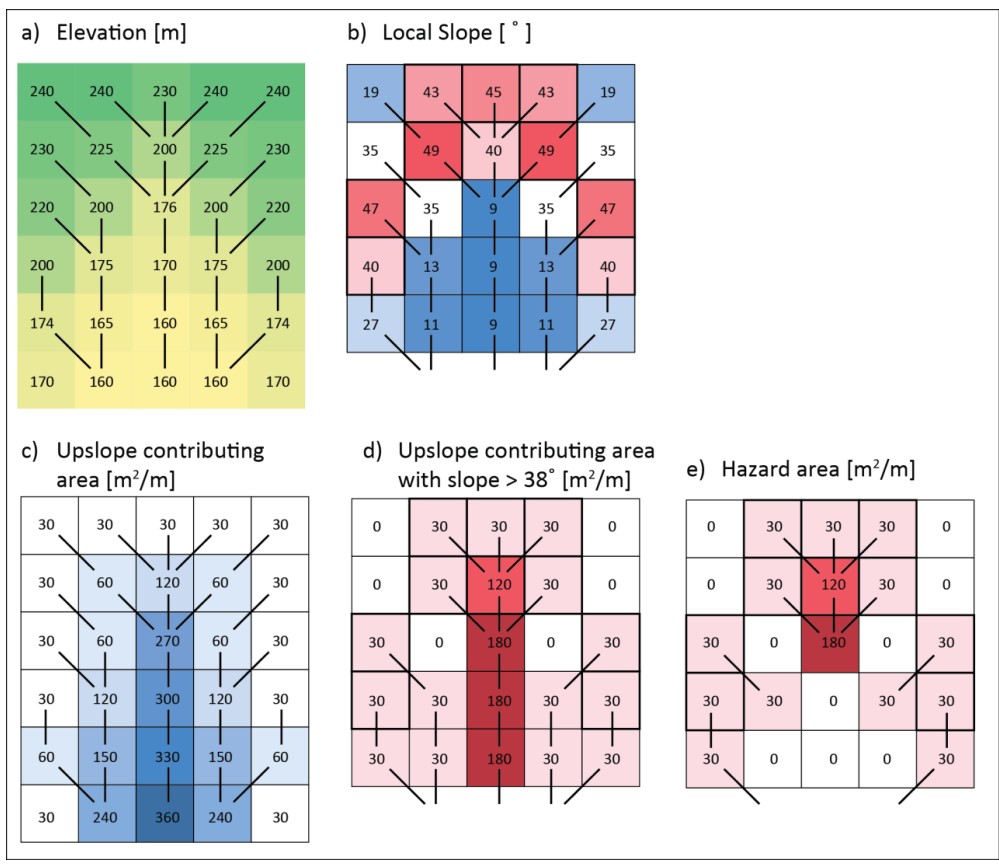


**Figure 2.** Worked example of SHALRUN-EQ hazard area calculations for an initiation angle of 39˚

and a stopping angle of 10˚. a), elevations from a 30 m resolution digital elevation model for an area

of topographic convergence. Lines show steepest flowpaths from cell to cell. b), local slope

calculated as the steepest path to a downslope neighbour. Thick outlines show cells steeper than

38˚. c), upslope contributing area using steepest flow path routing. d), upslope contributing area

steeper than 38˚. e), hazard area, defined as the upslope area steeper than 38˚ with flow paths that

do not fall below 10˚.




## 6. Results

### 6.1. Local slope



For all inventories, landslide probability increases as an approximately exponential function of local
slope (Figure 3a). For four of the six inventories, conditional probability exceeds the study area
average probability for slopes steeper than 30-35˚, with Northridge and Haiti lower at 20˚ and 25˚.
This suggests that slopes <30˚ are generally safer than average, while those >45˚ have a landslide
probability >200% of the average, and those >50˚ are generally >300% of the average. The curves
for Finisterre, Chi-Chi and Gorkha largely collapse on each other when normalised by study-area
average probability (Figure 3a). However, landslide hazard is less sensitive to slope for Wenchuan
and more sensitive for Northridge and Haiti. This variability between inventories likely reflects specific
study area properties such as the more dissected topography within the Northridge and Haiti study
areas. Comparing the amalgamated PDF of study area slopes (Figure 3a) with the conditional
probability curves indicates that the majority of the landslide hazard burden is held by the minority of
each study area (slopes >35˚). This implies that 1) many of the modest (<15˚) slopes on which people
generally choose to live are exposed to relatively low hazard (less than half the study area average
for all but Wenchuan); and 2) any choice to spend time or build infrastructure on steeper slopes
should recognise the considerable associated increase in exposure to coseismic landslide hazard.

### 6.2. Upslope contributing area


For all inventories, landslide probability increases from below the study area average at the lowest
upslope contributing areas – that is, ridge tops – to a peak or plateau at intermediate upslope
contributing areas, from which it declines in four of the six inventories (Figure 3b). Locations with the
lowest upslope contributing area also have the lowest landslide probability for four of the six
inventories, with Northridge and Finisterre as exceptions. For Northridge, the zone of lower than
average landslide probability extends only to upslope contributing areas ~40 $m^2/m$; for Finisterre it
extends to ~100 $m^2/m$, for Chi-Chi and Haiti to ~150 $m^2/m$ and for Wenchuan and Nepal to ~200
$m^2/m$. The location of peak landslide probability broadly coincides with the inflection in average slope
for a given upslope contributing area (Figure 4). This inflexion is commonly used as an indicator of
the transition from hillslopes to rivers (Montgomery and Foufoula-Georgiou, 1993; Stock and




Dietrich, 2006; Hancock and Evans, 2006), suggesting that maximum (or near maximum) landslide
probability occurs at the transition from hillslopes to channels (Figure 3b). Landslide probability
decreases with increasing upslope contributing area beyond this transition point for four of the six
inventories, gently for Finisterre and Chi-Chi, more steeply for Northridge and Haiti, and in all cases
with an increase in scatter that is likely due to the small number of observations with upslope
contributing area >1000 $m^2$/m.

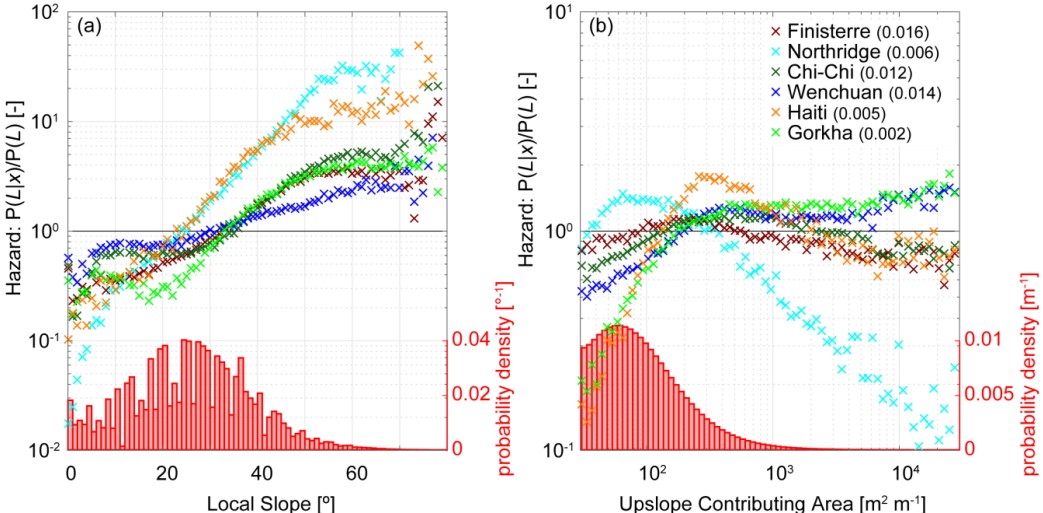


**Figure 3.** Landslide hazard defined as conditional probability P($L|x$) normalised by study area
average landslide probability P($L$), where $x$ is a) local slope and b) upslope contributing area per unit
contour length. Red bars show histograms of each variable over the six inventories. Note logarithmic
y-axes and different y-axis scales in panels a and b. The solid black lines show a normalised
probability of 1, equivalent to the study area average; thus, points above the solid black line have
conditional probability greater than the study area average. Legend includes study area average
landslide probabilities for each inventory (in brackets).

**6.3.       Local slope and upslope contributing area combined**
When slope and upslope contributing area are examined in combination, the highest landslide
probability is consistently found at the highest upslope contributing area for a given slope or the
highest slope for a given upslope contributing area (Figure 4). The lowest probabilities are found at
locations with both low slope and upslope contributing area, and cells with very low slopes have low





landslide probability almost independently of upslope contributing area. Importantly, landslide
probability increases more steeply with increasing slope than with increasing upslope contributing
area, indicating the dominance of local slope in setting landslide probability. This dominance is also
reflected in the orientation of the probability contours derived from logistic regression. There is
variability in contour orientations between inventories, with Finisterre and Northridge showing the
strongest slope dependence and Wenchuan showing the strongest upslope contributing area
dependence (Figure 4).

The shape of the two-dimensional probability surface determines the best course of action in terms
of choosing alternative locations for a particular asset or activity, but such action is also constrained
by what is possible. The average slope for each upslope contributing area (dashed line in Figure 4)
indicates that for Northridge, Finisterre, Chichi and Haiti there are rarely situations where a reduction
in upslope contributing area will not involve (on average) an increase in slope, that will actually
increase landslide probability. However, for locations in Wenchuan and Gorkha with upslope
contributing area of 300 to 10,000 m$^2$/m, the probability reduction due to reducing upslope
contributing area is not offset by the associated increase in slope. This suggests that, for the former
inventories, it is always beneficial to decrease slope even at the expense of upslope contributing
area, while for the latter it is more dependent on initial location. In general, the average slope contour
appears to separate higher and lower than average landslide probability in slope-upslope
contributing area space, suggesting that higher than average landslide probability is always found
on higher than average slopes for a given upslope contributing area.





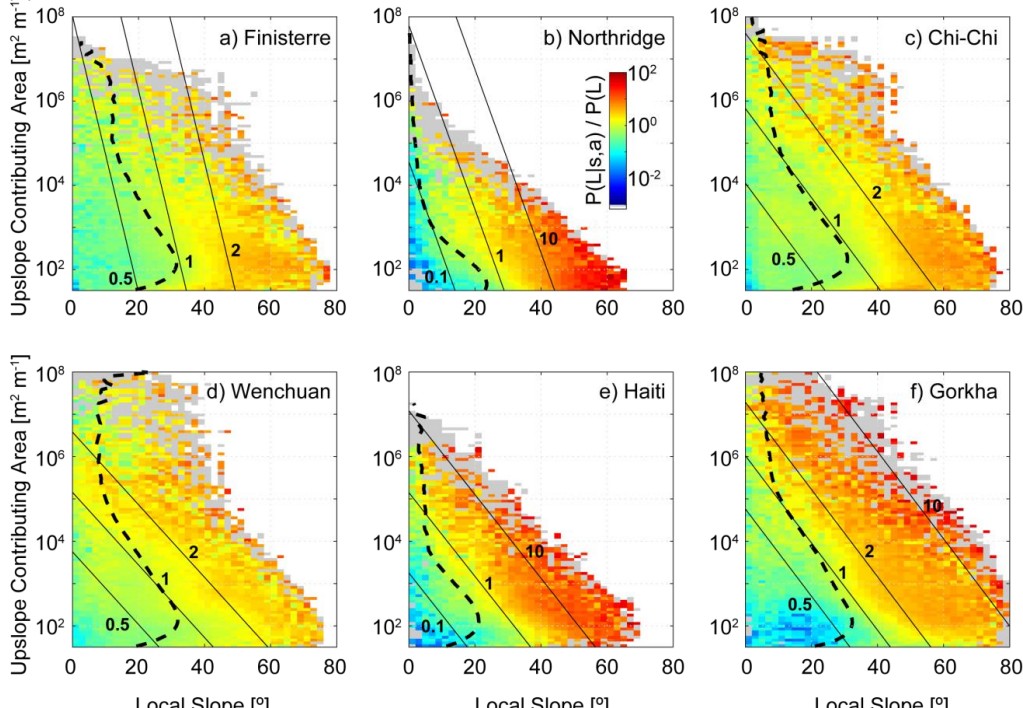


**Figure 4.** Two-dimensional plots of landslide hazard, defined as conditional landslide probability

P($L|s,a$) normalised by study area average landslide probability P($L$), where s is local slope and a is

upslope contributing area per unit contour length. Dashed lines show the mean slope per upslope

contributing area bin using 100 logarithmically-spaced bins. Solid lines are relative hazard contours

from logistic regression in the same units as the relative hazard surface. Grey cells indicate slope-

area pairs with data but with no cells touching a landslide. Note that upslope contributing area is

shown on a logarithmic axis, so that maintaining a constant landslide probability for a given increase

in slope requires a larger reduction in upslope contributing area at low slopes than at high slopes.

614



### 6.4. Skyline angle

Landslide probability increases as an approximately exponential function of maximum skyline angle (Figure 5a) as it does for local slope (Figure 3a). Landslide probability exceeds the study area average probability at skyline angles of 27-28˚ for Northridge and Haiti, 34˚ for Wenchuan and 38-40˚ for Finisterre, Chi-Chi and Gorkha. Locations with skyline angles of <20˚ have less than half the study area average landslide probability for all inventories, while those with skyline angles of >50˚ have more than double the study area average probability (Figure 5a). The lowest landslide probability values, at skyline angles of less than 10˚, are lower than those for local slope or upslope contributing area. As with local slope, the curves for several of the inventories (Finisterre, Chi-Chi and Wenchuan) collapse to a similar relationship when normalised by study area average probability suggesting similar behaviour across a range of different landscapes. However, Northrige and Haiti show stronger sensitivity to skyline angle and Gorkha shows considerably reduced landslide probability at low skyline angles relative to the other inventories.

### 6.5. Hazard area

The ability of hazard area $A_h$ to distinguish landslide from non-landslide cells is highly sensitive to two tuneable parameters ($\theta_m$ and $\theta_s$) but follows a smooth optimisation surface with a unique optimum for each inventory (Figure S1). Optimum parameters vary between inventories, with optimum initiation slopes $\theta_m$ ranging from 36˚ to 40˚ and stopping slopes $\theta_s$ from 6˚ to 31˚ (Table S1). Since these optimum parameters vary between inventories and can only be identified after an earthquake, they are problematic in terms of incorporation into a rule. Instead, we use the global average of the optimised parameter values from the six inventories ($\theta_m$ = 39˚ and $\theta_s$ 10˚). The stopping angle of 10˚ is steeper than many, though not all, of the observed slopes on which debris flows stop. For example, Stock and Dietrich (2003) report that debris-flow generally exhibit stopping angles of 2-6˚, but may halt at much larger angles (13-22˚) on open slopes. The steeper angles reported here, may reflect differences in the method and resolution of slope calculation but likely result from the coseismic trigger which does not necessitate high levels of saturation in the initial failure. Conditional probabilities are very low for cells with $A_h$ = 0 (i.e., where no cells steeper than





the initiation angle runout over flowpaths steeper than the stopping angle), ranging from 2% to 15%
of the study area average (Figure 5b). Conditional probability increases with $A_h$ for all inventories but
only slowly for $A_h < 1$ m$^2$/m; the trend then steepens to a peak (Northridge, Haiti, Nepal) or plateau
(Finisterre, Chichi, Wenchuan) at $A_h$ values of 100 to 1000 m$^2$/m with conditional probabilities 200 -
800% of the study area average (Figure 5b).

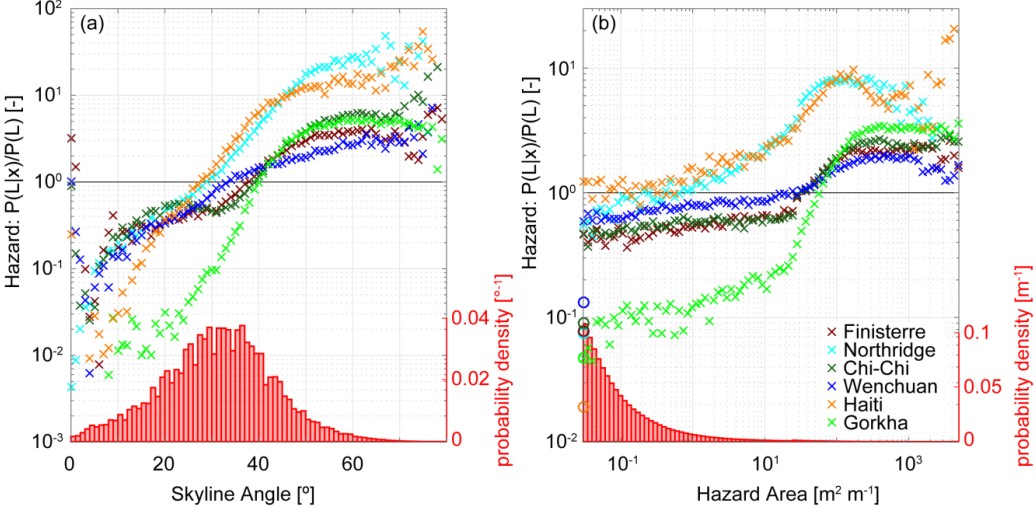


**Figure 5.** Landslide hazard defined as conditional landslide probability P($L|x$) normalised by study
area average landslide probability P($L$), for a) skyline angle; and b) hazard area with average
parameters - that is, the areas with slope greater than 39˚ that have a flow path to the cell of interest
and do not travel across a cell with a slope less than 10˚. Red bars show histograms of each variable
over the six inventories. Coloured circles on the y-axis in (b) indicate conditional probabilities for cells
with a hazard area of 0 m$^2$/m. Note logarithmic y-axes and different y-axis scales in panels a and b.
The solid black lines show a normalised probability of 1, equivalent to the study area average; thus,
points above the solid black line have conditional probability greater than the study area average.

**6.6.        ROC analysis**
To supplement conditional probability analysis, we examine the performance of slope, upslope
contributing area, skyline angle, and hazard area as continuous hazard indices (with high index
values reflecting high hazard and vice versa) using ROC curves (Figure 6). Successful indices will
capture landslide cells within high hazard index zones (true positives) without capturing non-

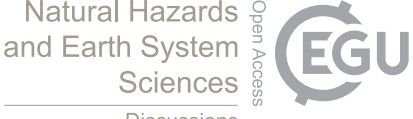

landslide cells in the same zones (false positives). Hazard area performs best for all six inventories
with an AUC always above 0.78 and an average AUC of 0.83 (Table 1). Skyline angle performs joint
best for Haiti and second best for a further three of the six inventories, with AUC always above 0.65
and an average AUC of 0.77. The exceptions, where slope, upslope area, or their combination
performs second best are Northridge and Wenchuan. For Northridge slope alone and slope plus
upslope contributing area both outperform skyline angle by a single percentage point, while upslope
contributing area by itself performs considerably worse (Figure 6a). For Wenchuan, upslope
contributing area considerably outperforms the other indices, perhaps reflecting longer-runout
landslides in this inventory, while slope performs particularly poorly (Figure 6d). Although slope,
upslope contributing area, and their combination all perform better than skyline angle in one of the
inventories. none do so consistently across multiple inventories. This is reflected in their averaged
AUC values over all inventories of 0.72, 0.72 and 0.73 for slope, upslope contributing area, and their
combination respectively.

**Table 1.** Area under the ROC curve for the five hazard metrics over the six coseismic landslide
inventories. The best performing metric for each inventory is in bold, the second best is in italics.

|  | Hazard area | Skyline angle | Slope + upslope contributing area | Local slope | Upslope contributing area |
|---|---|---|---|---|---|
| Finisterre | **0.79** | *0.72* | 0.69 | 0.69 | 0.66 |
| Northridge | **0.89** | 0.83 | *0.84* | *0.84* | 0.62 |
| Chi-Chi | **0.80** | *0.73* | 0.68 | 0.67 | 0.69 |
| Wenchuan | **0.78** | 0.65 | 0.62 | 0.58 | *0.74* |
| Haiti | **0.86** | *0.85* | 0.83 | 0.79 | 0.69 |
| Gorkha | **0.88** | *0.85* | 0.77 | 0.73 | 0.76 |
| Average | 0.83 | 0.77 | 0.74 | 0.72 | 0.69 |
| 1σ | 0.05 | 0.08 | 0.09 | 0.09 | 0.05 |




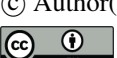


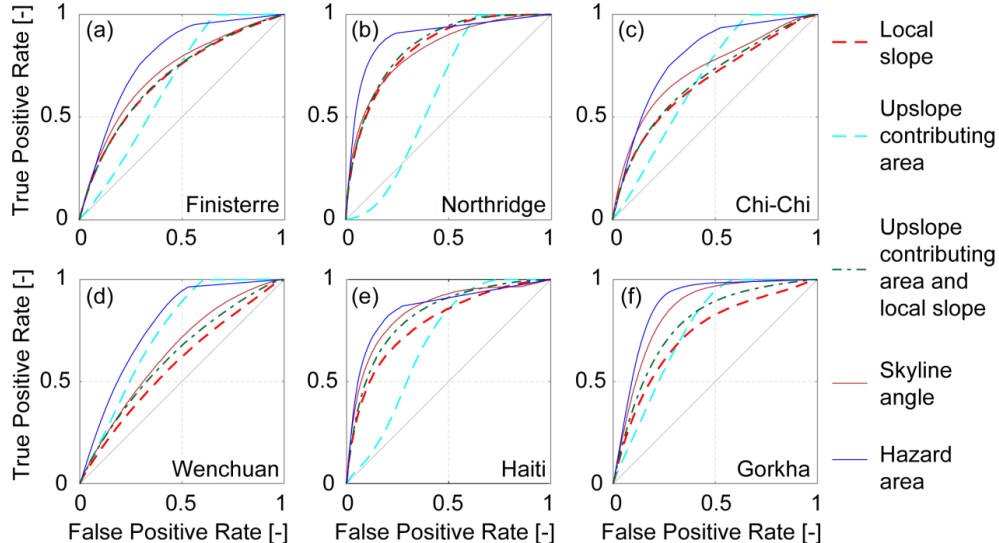


**Figure 6.** Receiver operating characteristic (ROC) curves for the six inventories: a) Finisterre, b) Northridge, c) Chi-Chi, d) Wenchuan, e) Haiti, f) Gorkha. False positive rate is given by the number of false positives divided by the sum of false positives and true negatives. True positive rate is given by the number of true positives divided by the sum of true positives and false negatives. The 1:1 line represents the naïve (random) case. Curves plotting closer to the top left corner of each panel represent better model performance.

### 7. Discussion

We structure the discussion around three simple rules that are drawn from the results above. In each case we explain the evidence on which the message is based, why it works, our degree of confidence, and implications for applying the rule. Finally, we examine the spatial implications of these rules using an example landscape.

#### 7.1. Rule 1: avoid steep (>10˚) channels with many steep (>39˚) areas that are upslope

The hazard area is the best or joint best predictor of landslide probability for all six inventories. The hazard area defined by the average initiation angle (39°) and stopping angle (10°) across all six inventories performs nearly as well as the optimised area for each inventory, enabling us to define a




general rule independent of any specific inventory. This is fortunate, as site-specific optimisation
requires a pre-existing landslide inventory for any individual area and so may not be feasible. In all
six inventories, locations with $A_h > 60$ m$^2$/m have landslide probability above the study area average.
While landslide probability generally increases with increasing hazard area, the relationship is
complex (Figure 6). Landslide hazard can be most effectively decreased by decreasing $A_h$ at
intermediate values of $A_h$, whereas decreasing $A_h$ at either the upper or lower extremes has minimal
effect on hazard. The qualitative statement to avoid areas with 'many' steep slopes could also be
phrased 'any' steep slopes since the landslide probability is generally 5-10 times higher even for
very small values of $A_h$ (c. 0.1 m$^2$/m) than the landslide probability for areas with no $A_h$.
Landslides do not always obey steepest flow path routing rules, and it is possible for landslides to
travel up reverse slopes or along contours. This is particularly true for large deep-seated landslides
or rockfalls. The hazard area metric cannot account for such behaviour and thus is more likely to
reflect hazard from smaller shallow landslides, while skyline angle, which does allow for runout over
reverse slopes, may be a better predictor for larger deep-seated landslides. The two indices have
some overlap but could be used in combination to find safer locations in the landscape.

**7.2.      Minimise your maximum angle to the skyline**
The maximum skyline angle is the second-best predictor of landslide probability in four of the six
cases. Locations with skyline angles less than 30˚ generally have a landslide probability below the
study area average. Importantly, landslide probability increases non-linearly with skyline angle, so
that a slight reduction to a high skyline angle results in a much larger reduction in landslide probability
than it would for a lower skyline angle.
The distinction between local slope and skyline angle reflects the importance of runout as well as
initiation in defining landslide hazard. Landslide hazard is an inherently non-local problem, defined
by both conditions at the point of interest and those upslope of that point. The skyline angle is a
simple way to represent this. It has the additional advantage of being easy to measure, needing only
a protractor or clinometer for precise measurement in the field, and being easily approximated by
eye. Local slope, in contrast, is scale-dependent, while upslope contributing area and $A_h$ are both
considerably more difficult to estimate in the field.

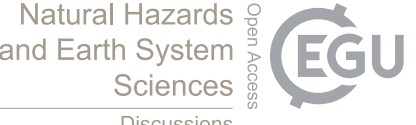




### 7.3. Minimise local slope, especially on steep slopes, and even at the expense of increasing upslope contributing area, but not at the expense of increasing skyline angle or hazard area

Local slope generally performs less well than skyline angle or hazard area but is a consistently skilful
predictor of coseismic landslide hazard, and could be a useful additional discriminant for situations
where both skyline angle and hazard area are comparable between two locations. In this situation,
our results suggest choosing the location with the lower local slope. This is particularly true at steeper
slopes since landslide probability increases exponentially with slope, approximately doubling for
every 10˚ increase in slope.
Given the common observation that coseismic landslides initiate near ridge crests (Densmore and
Hovius, 2000; Meunier et al., 2007), it is perhaps surprising that landslide hazard generally increases
with increasing upslope contributing area (i.e. moving downslope from ridge crests). In fact, while
coseismic landslides may initiate preferentially near the ridges, they runout downslope; thus, areas
near ridges are less likely to be touched by any part of a landslide even though they are more likely
than other parts of the landscape to contain a landslide crest. Landslide probability is consistently
low at very low values of upslope contributing area, corresponding to ridges; for some inventories, it
is also low at very high values of upslope contributing area, corresponding to valley floors in the
downstream reaches of the river network. This may be partly a function of the covariance between
local slope and area, since locations with large upslope contributing areas generally have lower
slopes (see dashed lines in Figure 4). The addition of upslope contributing area as a predictor in
logistic regression improves landslide probability prediction relative to slope alone (Table 1), but the
orientation of the probability contours (Figure 4) indicates that its influence is weak. Moving to a
location with lower slope angle almost always reduces landslide probability independently of the
upslope contributing area of the new location, although the specific reduction of landslide probability
depends on the shape of the two-dimensional probability surface (Figure 4). We conclude that
decisions on how to reduce landslide hazard most effectively need to be made on a case by case
basis, and are best made using hazard area, skyline angle, and the local slope in conjunction with
each other. Steep upslope areas result in elevated hazard but gentle upslope areas do not,





explaining the improved performance of hazard area relative to upslope contributing area (Figure 6
and Table 1). Ridges, with very low upslope contributing area, are generally low hazard locations if
they have gentle local slope but can still be hazardous if they are steep (Figure 4). To minimise
landslide hazard, it is thus preferable to seek broad ridges over sharp ridges where such a choice is
possible.

**7.4.    Movement rules in a landscape with variable hazard**
While this analysis is focused on cell-by-cell hazard assessment, and is thus appropriate for
decision-making before a large earthquake, it is also possible to use the results to define some rules
for movement or relocation during or immediately after an earthquake. Our analysis shows that even
during a large earthquake in mountainous terrain, landslide hazard is not ubiquitously high. A
significant fraction of the landscape has low landslide probability (<5% of the study area average) –
as much as 30% in Northridge and 33% in Nepal. This means that it is often possible to find locations
with lower landslide hazard. Landslide hazard is extremely granular in spatial terms, so that small
changes in location can make a big difference to exposure. The vast majority of locations (75% in
Nepal, 95% in Northridge) are within 1 km of areas of low landslide probability (<5% of the study
area average). Even smaller movements of 100 m or less, as might be possible during or immediately
after a large earthquake, can result in very large reductions in hazard.
Detailed analysis in the Northridge (Figure 7) and Nepal inventories shows that landslide hazard can
often be effectively reduced by moving from a slope to a ridge (e.g., from A to B in Figure 7), out of
a gully (e.g., from C to D), or downstream of a flatter area (e.g., from C to E). However, there is no
single answer to the question of where to move to reduce coseismic landslide hazard, since this
differs depending on the setting, the distance that can be travelled due to time or location constraints,
and on the chosen rule (e.g., skyline angle vs. hazard area). Given a 1 km radius of potential
movement, minimizing skyline angle involves moving upslope for ~75% of locations in Nepal but
only ~66% in Northridge. In some cases, knowing how far one can travel can be critical: if one may
only travel a short distance, moving upslope may be preferable (e.g., from C to D in Figure 7), while
if one could travel farther, moving downslope may offer greater hazard reduction (e.g., from C to F
or G).





Landslide probability estimates for high hazard locations are broadly comparable between skyline
angle and hazard area metrics (e.g. Figure 7). However, different metrics emphasise different parts
of the landscape. Ridges consistently minimise skyline angle but may still have intermediate values
of hazard area if the ridge is sharp so that the local slope of the ridge itself is steep. Broad valley
floors consistently minimise hazard area, but may still have intermediate values of skyline angle if
the neighbouring slopes have sufficient relief. There are trade-offs between these metrics, and
further work is needed into how they might be combined to further reduce hazard.

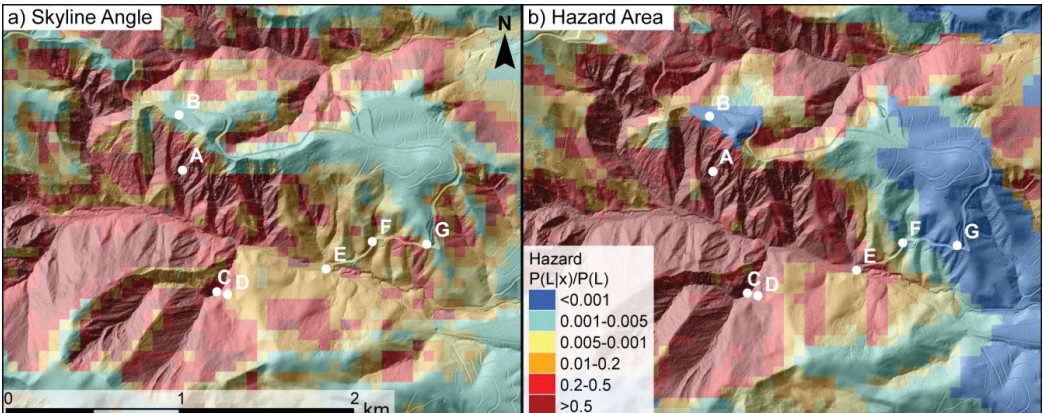


**Figure 7.** Example landslide hazard estimates derived from a) skyline angle and b) hazard area for
a small section of the Northridge study area. Colours reflect landslide hazard estimated from the
two methods, expressed as a fraction of the study area average hazard. Points labelled A-G in
white are example locations discussed in Section 7.4. Hazard estimates are overlain on shaded
relief from a 0.5 m resolution LiDAR DEM for context (source: NCALM, 2015,
DOI:10.5069/G9TB14V2).

**7.5 Caveats**
These rules should be combined with existing guidance, such as local knowledge and formal hazard
and risk information when that is available. The rules provide an evidence base that could be used,
for example, in infrastructure and land-use planning, identifying evacuation routes, and designing
contingency plans from individual to community level, where more detailed or formal technical advice
is not available. It is also important to note some caveats.



This analysis is purely focussed on coseismic landslide *hazard*, and thus it does not take into account
the distribution of vulnerability: that is, the locations of people and infrastructure in these landscapes
or how they might be differentially impacted by landslides. While one area may be more hazardous
than another, the distribution of people and infrastructure may be such that risk is not actually
increased. Further, our analysis is probabilistic, defining hazard as the probability of intersecting a
landslide; thus, our rules identify locations where the landslide probability is lower, not where
probability is zero. This means that it is possible for an alternate location chosen based on its lower
landslide probability to be impacted by a landslide while the original higher-probability location is not.
The choice of inventory will influence the specific results and, although we adjust for bulk shaking
intensity by normalising conditional probability by bulk probability, differences between inventories
are likely to remain (e.g., in spatial patterns of shaking intensity and their relation to topography).
Rock type is a critical influence on landslide occurrence (Chen et al., 2012; Harp et al., 2016; Roback
et al., 2018), but we have excluded it from our analysis because it is extremely difficult for an
untrained observer to identify and to translate into meaningful estimates of material strength and
thus landslide probability. While rock type is likely to influence the relationship between topography
and landslide hazard (e.g., Chen et al., 2012) we expect the length scales over which this occurs to
be long (order kilometres) relative to the other factors examined here.
Because the analysis is focussed on *coseismic landslide* hazard, it does not account for other
sources of hazard, either associated with an earthquake (e.g., seismic amplification on ridges), or
with other processes or events such as flooding. In some cases, following our rules in isolation might
increase exposure to other hazards. For example, moving to ridge tops to minimise skyline angle
might increase exposure to intense shaking due to seismic amplification; moving to valley floors that
are occupied by large rivers, where hazard area is minimal, might increase exposure to fluvial
flooding. We also have not considered the effects of landslide size or failure type, choosing instead
to treat all landslides as representing an equivalent hazard. If landslide size or type shows a strong
spatial dependence, then parts of the landscape may be preferentially impacted in ways that are not
reflected by our rules. Finally, it is not yet clear how transferrable our conditional probability results
are to rainfall-triggered landslides. For instance, stopping angles are likely to be lower for rainfall-
triggered landslides where the failing mass is more highly saturated (e.g. Stock and Dietrich, 2003).





Similarly, in the case of rainfall-triggered landslides, initiation is likely to depend not only on slope
angle but also topographic control on saturation (e.g. Bellugi et al., 2011). Extending the analysis to
other triggering mechanisms is thus a future research need.

**8. Conclusions**
We have introduced a set of simple rules that can be used to identify, and thus potentially reduce,
exposure to earthquake-triggered landslides. We test a set of candidate predictors for their ability to
reproduce mapped landslide distributions from six recent earthquakes. Landslide hazard, defined as
the conditional probability of intersecting a landslide in one of the six earthquakes, increases
exponentially with local slope. Landslide hazard on hillslopes also increases with upslope
contributing area, suggesting that while ridges may be areas of preferential coseismic landslide
initiation, they are not the locations of highest coseismic landslide hazard due to downslope
movement of landslide material during runout. When accounting for both slope and upslope
contributing area, landslide hazard is highest for the highest area at a given slope or the highest
slope at a given area. Landslide hazard can be reduced by reducing local slope, even at the cost of
increased upslope contributing area, and especially at high slopes. Landslide hazard increases
exponentially with the skyline angle, and this simple, easily-measured, metric performs better than
slope or upslope contributing area for four of the six inventories. Hazard area, which accounts for
both landslide initiation and runout, offers the best predictive skill for all six inventories but is more
difficult to estimate in the field and requires estimation of two empirical parameters. Fortunately,
hazard area calculated with parameters that are averaged across all six study sites (initiation angle
of 39˚ and stopping angle of 10˚) performs only slightly worse than hazard area calculated with
optimised site-specific parameters, suggesting that the average parameters can be applied to other
inventories. These findings can be distilled into three simple rules:
1) Avoid steep (>10˚) channels with many steep (>39˚) areas that are upslope;
2) Minimise your maximum angle to the skyline; and
3) Minimise local slope, especially on steep slopes and even at the expense of increasing

863       upslope contributing area, but not at the expense of increasing skyline angle or hazard area.




**Acknowledgements**
This work was financially supported by grants from the NERC/ESRC Increasing Resilience to
Natural Hazards programme (NE/J01995X/1) and the NERC/ESRC/NNSFC Increasing Resilience
to Natural Hazards in China programme (NE/N012216/1). We thank: 1) colleagues at the National
Society for Earthquake Technology-Nepal (NSET) who have helped to shape our thinking on
landslide hazard and the challenge of risk communication; 2) those responsible for collecting the
landslide inventories used in this study, particularly Neils Hovius and those that contributed their
data to the ScienceBase-Catalog; and 3) William Dietrich and Neils Hovius for helpful comments
on an earlier draft. LiDAR data acquisition and processing completed by the National Center for
Airborne Laser Mapping (NCALM). NCALM funding provided by NSF's Division of Earth Sciences,
Instrumentation and Facilities Program. EAR-1043051.

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
