# Peer review of "Simple rules to minimize exposure to coseismic landslide hazard"

_Natural Hazards and Earth System Sciences, 2018_

## Short Comment (SC1) · 23 Nov 2018

Dear authors,

I enjoyed reading your manuscript, which I believe can be a useful contribution towards landslide risk reduction in highly seismic regions.

I have a few questions, mostly regarding the robustness of your findings, which I list as follows:

- You mentioned multiple times that the DEM resolution can influence some of your results. It would be nice to quantify this influence at least for one inventory for which a higher resolution DEM is available (e.g. Northridge). Perhaps, moving from 30 m to 10 m DEM will only produce marginal improvements while increasing the computational

[Figure]

cost significantly, or on the contrary it will change the result significantly.

- There are cases in which several inventories are available for the same study area (e.g. Wenchuan). These inventories are sometimes quite different from each other. Among others, we discussed this in a recent submission, still under review (see the revised manuscript in the discussion at https://www.earth-syst-sci-data-discuss.net/essd-2018-105/) and we found substantial areal mismatches (up to 67%) between inventories in the Wenchuan, and rather low pixel-based correlations (R-squared as low as 0.35). We showed that this translates in quite some differences in landslide-size probability distributions and hence in landslide volume estimations. This might condition some types of hazard assessments based on volume-runout correlations. However, we did not go deeper into the topic, as it was out of the scope of our manuscript, and we did not investigate how this mismatch between inventories translates into statistics of controlling factors (e.g. slope, upstream contributing area, etc.). It would be interesting if you could estimate to what extent choosing a different inventory for the same study area would affect your assessment.

- Also, again about the Wenchuan case, you only chose a subset of the inventory by Li et al. (2014) containing about 1/3 of the landslides. It would be good to explain whether this subset can be thought as representative of the entire study area (e.g. in terms of landslide metrics, topography, lithology, distance from epicentre and fault rupture, etc.) so that one would be confident that the results you obtain have more general validity and are not biased by your choice, which was only due to a data availability issue. What you report in the conclusion (see my point below), that is that the site-specific and averaged rules perform similarly, is comforting in this sense, but what if it is just a coincidence?

- From your analyses you obtained a set of simple and easily understandable rules to minimise the exposure, and you wrote that the hazard area calculated with averaged parameters performs only slightly worse than hazard area calculated with site-specific parameters. This is encouraging and, as you wrote, it suggests that the average parameters can be applied to other inventories (or subsets of inventories). Thus, it would be very interesting to see these averaged parameters being applied to other inventories, across a variety of landscapes, climates and seismic characteristics. Also, it would be interesting to apply your rules to a highly seismic region in which no recent earthquake has occurred, and relate it to the current distribution of population and exposed goods (but I recognise the latter is out of the scope of this work, so it is just an idea).

Thanks again for your excellent contribution.

Gianvito Scaringi

---

## Referee Comment (RC1) · Marc (Referee) · 2 Dec 2018

SUMMARY

Milledge et al., present a thorough statistical analysis of six coseismic landslides inventories to relate landslide hazard to landscape properties such as slope and contributing areas, but also more specific variables such as skyline angle and a hazard area integrating the probability of initiation and propagation of landslides.
They found that the two latter metrics explain best the location of the inventories and may allow to be converted into simple rules useful for hazard management.
The paper is well written, with a straight forward structure and informative. It will make a nice contribution for NHESS both for its systematic analysis and its recommendations.

I have two major comments that I think could improve the results and the discussion, and then give a number of minor Line by Line comments with potential clarification or additional small analysis.

MAJOR COMMENTS

My first comment is about the normalization of several of the hazard metrics : I am convinced that a substantial part of the difference between the hazard curves could be removed by plotting the hazard against a landscape metric : For example for slope, each landscape as likely a modal slope, that may be interpreted as the result of geolechanical difference (for steady state landscape at least). Thus curves may be plotted against S-mode(S) , somewhat normalizing for difference between two landscape.  I can understand the author may still want to express their rules in terms of absolute values of slope or other variables, but I suspect this normalization would clarify and strengthen the result and their analysis (as this did in other studies). I make suggestion for the other variables in my inline comments.

My second concern is that their maybe some over-interpretation of the data scatter towards the extremity of the hazard curves. And the author do not provide clear metrics or indication of the validity of individual datapoint. This is not an easy task but the work of Rault et al., which I co-authored, recently proposed a method to do exactly that. I would suggest the author to apply these criterium and check.
 In this work we consider the probabilty p of the whole topography, and the one resulting from the landslides affected area only $p_L$. To assess whether $p_L$ is significantly different from p we compute the confidence interval Ip associated to the random drawing of n (n the number of landslides) pixels out of the landscape distribution. If $p_L$ belongs to [p-Ip : p+Ip] then we cannot exclude that the difference between p and $p_L$ just comes from random fluctuations and it is likely not significant. Given landslides remain rare in the whole topography, the drawing can be assumed independent, and similar to a Bernoulli sampling. Provided the central limit theorem is respected (i.e. **n>30, np>5 and n(1-p)>5 )** the 90% confidence interval can be estimated as :
**Ip** = p – 1.96 (p(1-p)/n)^0.5 ;  p + 1.96 (p(1-p)/n)^0.5.  Some additional details can be found in the supplementary methods of Rault et al., 2018.
Basically n is large (n>1000-10,000) so the authors should obtain very narrow Ip until they reach p<0.001 – 0.0001 but I expect these low probability to be reached in the tail of the distribution (Fig 3,4, 6) and the cut off will vary for the different landscape with higher or lower p or n. The authors could compute **Ip as well as the convergence**

**criterium and show the points which may be insignificant in shaded / transparent ?**

**Line By Line comments:**

L123 : I could not find Milledge 2018 in the reference list... please check.

L133: Add couple of reference for shaking: e.g. Khazai and Sitar 2004, Meunier 2007.

L138: I think you should also cite Meunier 2008 here, and probably the recent analysis discussion for an extended number of earthquakes in Rault et al., 2018

L142:152 : A couple of references on the suspected effects would be relevant. Especially the ones cited elsewhere in the text : Parise and Jibson 2000, for lithology , Maufroy et al., 2015 for curvature and ridge amplification.

L155: True they pertain to initiation, but vast majority of studies highlighting their role or quantifying statistical relations between these predictors and landslide use total area and therefore are combining both initiation and runout.

L180: I have the impression it should be the minimum skyline angle, not intersecting topography, …. Indeed a maximum reach angle.  Cf comment on Fig 1

L200 – 300 : This is certainly at the appreciation of the authors, but I have the impression the earthquake environment (tectonic, climatic, vegetation) is over described. Given you never re-refer to this context later, you may shrink those description and end this section with a sentence like : "these epicentral areas encompasses a large diversity of tectonic (X to Z) , climatic (X to Y) and vegetation cover ( X to Y) contexts, but we assume landslides in all of them should be at first order driven by topographic parameters in the same way".
In contrast, some aspects may be missing or insufficiently discussed:
1/ I think the number of landslide polygons used in Chi-Chi is missing.
2/ The fact you used a sub-inventories in Wenchuan may mean you artificially limit your analysis to a range of shaking quite different from the other cases . This should be mentionned.
3/ A few words on the Implications of polygon mapping quality on your analysis may be given here (or in discussion), such as the affect of amalgamation ; inclusion or not of debris flow propagation within the river network ? (I know it is a difficult distinction, often ignored but it should impact the statistics, especially of contributing area for example). Implictions of the different resolution limit  (for Chi Chi or FInisterre compared to Northridge ) or of the location accuracy ?

Figure 1: Caption: a cone projected from P no ?

If your cone as angle define from horizontal upward, are you looking for the minimum skyline angle, not intersecting the topography? That is what I get from your sketch in c). See previous comment about skyline angle definition.

L372-380: < 10 observations ? Why ? And this is only for the 2 ariable case (Fig 4). For the single variable case it is not clear what is noisy data and where to really set the boundary or unsignificant datapoints. Rault et al., 2018 propose an extension of Meunier et al., 2008 studies with an estimation of the uncertainty of observations based on both the number of observations and the probability. See major comments.

L458: Shalrun-EQ= Probability of mobilization convolved with connection probability. Average in the above area. So hazard area is basically the number of pixel where debris flow can occur and reach the interest cell (say Nhaz)... in the contributing area, times pixel area, and divided by the contour length, i.e. the square root of contributing area.
Although I am confused because in Fig 2 : Hazard Area seem to be Nhaz times Pixel resolution (or Nhaz.a/sqrt(a) ). But then smallest vales should be 0 and 30 (as it does in Fig 2). But in Fig 6 it goes from 0.1 to 1e3… So there seems to be a problem between the 2 definitions. Please check.

L462: repeat from L452-453. Cut or rephrase ?

L478-482 : Steepest descent may be too conservative, even if your rule needs to be simple, maybe you could mention that probability to propagate on non-steepest descent path is probably non-null.
Also what about landslide large enough to be continuing beyond the first cell with angle below the deposition threshold ?
I see this is partially acknowledged in the discussion. Maybe you can flag here the fact you discuss suc limts later.

L512-516 : Ok, simplicity is important and it is difficult to integrate other effect mentioned here. But what about checking the actual evolution of probability with slope, for both initiation and stop ?
A reasonable estimate of scar area can be obtained by selecting the highest elevation pixel in your landslide, and selecting as many as needed to reach a scar area with an aspect-ratio of 1.5 (Domej et al., 2017) and a mean width representing of your polygon (see Marc et al., 2018 for how to do that). Doing so you could check if a plateau develop in your probability ratio after 39° or so... Interestingly you could reverse the idea and take the lowest N pixel (N ~ Width / 30 for 30m resolution DEM) of your landslides to obtain a probability of stopping.

L536 : Did you check the curve appearance when using gradients, that is tan(Theta) (with theta the slope in degree) ? Because the tan(theta) does appear mostly exponential over a large range of Theta. Thus a linear function of tan(theta) (the

relevant parameter for landslide stability) may appear exponential when plotted against theta.

L538-542 : Northridge and Haiti are shifted compared to other. They both become > average probability around 20° , vs 30 for others. This roughly correspond to modal slopes of these areas.
It would be interesting to re-plot all curves not against slope, but slope – Sm the modal slopes. This collapsed curves on a similar analysis for rainfall (cf Marc et al 2018)

Similarly is there a large variety of drainage area distribution ? Haiti and Northridge are very peculiar again compared to the other cases. Some normalization by the mode of the landscape drainage area may be important.

L542-543: If you consider that Haiti and Northridge are more sensitive because they reach higher ratio it may be a confusion because of the lack of normalization (previous comment). It is plausible the relation between slope – Sm and hazard is similar, only the difference between resolvable slope (with a 30m DEM) and the modal slope is larger, allowing to reach larger relative hazard. I think the effect of normalizing for the landscape must be assessed.

L545: combined or merged PDF rather than amalgamated (that sounds negative an unusual to me but I may be wrong).

L555: You say you observe contributing area, but you have normalized by contour length. In the paragraph about hazard (L489), you say contour length is $a^{0.5}$, but it is not so clear what is a (the area of a cell, which cell ?) On Fig 2, contributing area seems to be the square root of $a$. It would be consistent with the contour length estimated as sqrt(a) but then why not say straight you look at the sqrt of drainage area ? Maybe I missed something, or it is worth clarifying a bit.

L562: This was somehow my expectation, so why not normalizing the contributing area and thus analyzing $a/a\_rc$ , with a_rc the channel ridge transition area? Like this the relative decrease or increase away from this objective characterization of the landscape could be analyzed (and the plot in Fig 3,4 would compare hazard curve shape only, not locations). This seems like an important improvement even if I understand that you may point to the fact a layman user of an hazard rule may not guess the modal slope of its landscape or the value of a_rc. After some analysis in the normalized domain general rules for the natural domain may be derived.

Fig 4 is very interesting and make a lot of sense after Fig 3.
However, I am wondering about two things…
1/ Would all the plot look the same if you use normalized area and slope ? Maybe not given that it was not expected from Fig 3 that Finisterre would be different, but it seems worth and easy to check.
2/ You work with 100 log-bins of a and it seems 1degree bins of slope. So I wonder what is your typical number of DEM cells in each of your bins, and thus how statistically

significant bins are… This is a detail as pattern are very consistent and a larger bin size would rapidly increase the amount of data.

Fig 5 : Skyline angle is strongly uni-modal. So I would study all areas with a relative skyline hazard: Sky- Modal(Sky). The modal will account for difference in incision/relief between landscape.  A potential outcome of such normalization may be that your case have all similar behavior for high skyline angle (increase and then plateau) but that Gorkha, Haiti, Northridge have a steep decrease below a certain angle while not the three others.
The definition you take for hazard area gives 0 hazard area for the reference in all cases and then a decrease. It does not seem that shift in the horizontal direction would do any good, and the vertical shift seems due to the proportion of zero hazard area in the landscape, so maybe computing a landscape PDF ignoring the zero would be insightful ?

L645: $Ah<1$ m²/m . I am surprised by this threshold, but maybe it is a typo. I would have say in Fig 5b the curves steepens most in all case around 20.
It is true that for Haiti, Gorkha and Northridge there is a slight increase in the trend after $Ah\sim1$, but minor compare to the later steepening.
Also to be sure that the difference between a peak or  a plateau is a real result it would be important to check the evolution of the uncertainty in your last bins, where certainly few data are available (even if we cannot read the probability of $Ah>1e2$ or $1e3$).
 We also do not see the difference in availability of such high hazard area in the different areas, so could a very low availability of such hazard areas in Haiti and Northridge (that have less steep slopes) caused a scattered behavior for $Ah>100$ instead of $Ah>1000$. A quantification of uncertainty may clarify that. See major comment.

L672 : This sentence confused me. Do you mean each of the three parameters, may be better than the skyline angle for at least one event ?

L674: These values do not match Table 1 with 0.72, 0.69, 0.74.... Please correct one or the other.

L694: I am a bit surprise by the term of channel inside this rule. I guess it derives from the fact that the hazard consider upslope contributing areas defined from flow algorithm. But the hazard area at many intermediate locations on hillslopes may be a channel for your analysis but not for the resident and deciders of the area. Because a channel is defined on finer scale than the DEM. You already say that this metrics is anyway difficult to estimate and handle for application, but this terminology would also complexify the problem for deciders or policy makers.

L699 : This is fortunate indeed, almost surprising.

L711: Interesting. Do you think this could be somewhat validated by making skyline and hazard graph for landslide above and below a certain threshold (say 5e3 m2 or even better above a certain width...) ?

L739 : You certainly mean Meunier 2008 here. However, note that the new study from Rault et al., 2018 is considerably nuancing these past studies.

L820: And even for a trained observer.
L822-23: I do not understand what you mean by "we expect the length scales over which this occurs to be long (order kilometres) relative to the other factors examined here"
Do you mean that main lithological units are usually big (regional scales) and thus significant part of a landscape will have homogeneous lithology, whereas topographic attribute change at the scales of 10s of meter ? Then it is the lengthscale for the variability of lithology that you want to mention. Anyway please clarify.

On a side comment, normalizing each landscape slope by their modal slope would be somehow a step toward normalizing difference in landscape that can be due to major lithological or geomechanical attributes (Korup 2008).

L824-826: This is an important and natural point to make but I would mention rainfall induced landslides straight here, as area affected by coseismic landslides are often even more often affected by rainfall induced landslides ( at least for wet climate Nepal, Finisterre, Taiwan).

L830: And they likely do, given that large landslide (likely to travel further away as you recall in the introduction) are usually reported closer of the fault or at larger shaking values (Khazai and Sitar (2004), for the Chi-Chi earthquake (1999), Massey et al. (2018) for Kaikoura or Valagussa et al 2019 for systematic evaluation of PGA and landslide size distribution.
So future exploration of the behavior of your hazard curve split for specific lithology of different area class should be done.

L834 : I would say we can reasonably expect strong differences : given that hazard increase strongly with local slope for EQ (Fig 4) but not for the rainfall induced landslides : as shown by the anaysis similar to your Fig 3 in Marc et al., 2018. Further, the longer runout (due to lower stopping angles) and stronger dependence on contributing areas are additional changes.

**References used in the review**

**Claire Rault, Alexandra Robert, Odin Marc, Niels Hovius, and Patrick Meunier,** Seismic and geologic controls on spatial clustering of landslides in three large earthquakes, 2018, https://www.earth-surf-dynam-discuss.net/esurf-2018-82/

Marc, O., Stumpf, A., Malet, J. P., Gosset, M., Uchida, T. and Chiang, S. H.: Towards a global database of rainfall-induced landslide inventories: first insights from past and new events, Earth Surface Dynamics

Discussions, (March), 1–28, doi:10.5194/esurf-2018-20, 2018.

Maufroy, E., Cruz-Atienza, V. M., Cotton, F. and Gaffet, S.: Frequency-Scaled Curvature as a Proxy for Topographic SiteEffect Amplification and Ground-Motion Variability, Bulletin of the seismological society of America, 105(1), 354–367, 2015

Valagussa, A., Marc, O., Frattini, P., and Crosta, G. B.: Seismic and geologic controls on earthquake-induced landslide size, Earth and Planetary Science Letters, 2019. https://doi.org/10.1016/j.epsl.2018.11.005

Bijan Khazai, Nicholas Sitar, Evaluation of factors controlling earthquake-induced landslides caused by Chi-Chi earthquake and comparison with the Northridge and Loma Prieta events, Engineering Geology, 2004, https://doi.org/10.1016/S0013-7952(03)00127-3.

Domej, G., Bourdeau, C., and Lenti, L.: Mean Landslide Geometries Inferred from a Global Database of Earthquakeand Non-Earthquake-Triggered Landslides, Italian Journal of Engineering Geology and Environment, 87–107, https://doi.org/10.4408/IJEGE.2017-02.O-05, 2017

Korup, O. (2008), Rock type leaves topographic signature in landslide-dominated mountain ranges, *Geophys. Res. Lett.*, 35, L11402, doi: 10.1029/2008GL034157.

---

## Referee Comment (RC2) · Anonymous Referee #2 · 3 Dec 2018

General comments

Thank you for this interesting paper. Using six inventories of coseismic landslides, the authors test the significance of multiple topographical parameters to constrain a set of simple rules in order to minimise exposure to landslide hazard. The paper forms a significant added value to the landslide hazard scientific community as a first attempt in identifying simple rules which is essential for communication about complex hazards to a broad (lay) audience in creating awareness and minimizing landslide exposure.

I appreciate the authors' balanced conclusion on the most effective parameters for hazard reduction ["We conclude that decisions on how to reduce landslide hazard most effectively need to be made on a case by case basis, and are best made using hazard area, skyline angle, and the local slope in conjunction with each other."], unfortunately

this is not taken in the abstract and conclusion where the authors present without further nuances three simple rules. The discussion is focused on the authors' results with limited reflections with respect to related research (cf. introduction). I believe such a reflection would make the results more convincing.

Specific comments

- The first time I read through the paper I found the abstract and introduction confusing while the terms hazard, exposure, risk, hazard response, "anticipating". . . are used without first clearly constraining them. Even though the audience from NHESS should be familiar with these terms I believe that these terms are still easily confused. I would therefore recommend to distinguish these terms in the introduction, or make reference to literature in which this is done.

- The paper is well structured and the figures of high quality presenting very clearly the results, yet I would suggest to shorten the paper to bring forward the main messages even more clearly. Sections that I would suggest to reduce are section 4 ("Earthquake inventories") by providing a summary of the used inventories with the most important parameters necessary for the analysis; and section 5 ("Methods") could also be reduced, moreover this would allow the reader to more easily follow the workflow.

- I wonder how easily the presented rules can be adopted without prior knowledge or skills, which seems to be the main purpose of the study yet lacking from the discussion. This is not easily answered and out of scope of the study to check the applicability of their rules by householders, local government, and NGOs, but I would recommend to be more cautious when claiming to present 'simple rules'.

Please also note the supplement to this comment:
https://www.nat-hazards-earth-syst-sci-discuss.net/nhess-2018-271/nhess-2018-271-RC2-supplement.pdf

2018-271, 2018.

**Supplement:**

**Detailed comments on "Simple rules to minimize exposure to coseismic landslide hazard"**

L10 - The abstract misses information on the fact that the study is on coseismic landslide hazard.

L15 - Do you present in the end primarily simple rules to identify hazard? Or rules to minimize exposure, cf title? I understand they go hand in hand, but it would be good in my opinion to be aware that the terms Hazard, Exposure and Risk are easily confused by readers. Being consequent in using terminology in the abstract might avoid confusion.

L18 - Not sure what you mean with "as a proxy for hillslope location".

L20 - From reading only the abstract it is difficult to agree that defining "the upslope area with slope >39° that reaches a location without passing over a slope of <10°" does not require prior knowledge or skills and that it is easy understandable.

L22 - Could you add the observation period covered by the inventories here between brackets to know what is 'recent' to you?

L23 - Show which other metrics were tested besides the two new metrics you introduce so this sentence ("most skilful") has more meaning.

L25 - If the rules should be simple and applied by people without skills, why not round to 40°? What is the sensitivity of this rule to a change in the slope of one degree?

L26 - How does that work, "minimise local slope especially on steep slopes"?

L26-28 - This rule seems dubious when stating at the same time "even at the expense of increasing upslope contributing area" and " but not at the expense of […] hazard area" with the latter also comprising upslope contributing area.

L38 - I would suggest to use the updated paper of Petley, 2012:
Froude, Melanie J., and D. Petley. "Global fatal landslide occurrence from 2004 to 2016." Natural Hazards and Earth System Sciences 18 (2018): 2161-2181. Given the very extensive reference list I think that 'e.g. Froude et al. 2018' would do while omitting the other references if not necessary in the rest of the paper.

L46 - I think "respond to that hazard" is of lesser relevance here as you do not deal with hazard response in this paper.

L55 - I would add to "site-specific information that may not be available" something like "such as… " to make it more informative.

L62 – "hazard maps cannot resolve hazard at those scales" : I doubt that, with the current availability of high-resolution remote sensing data; yet I agree it could be time-consuming.

L97 - How does the "self-recovery" relate to the first part of the sentence? I don't see the relevance of it here.

L102 - Not only of "less use" but also inherently different; your rules aim to minimize landslide exposure, not to help in hazard response. Please modify.

L107 - Could you site a reference at the end of this sentence, in order to make "our" refer to the scientific background.

L110 - Add respective countries between brackets.

L112-116 - I don't see much difference between the two questions?

L116 - What kind of patterns? Temporal/spatial...

L118 - Which "combined datasets" you refer to? The landslide inventories or more specifically to the derived topographical parameters from the inventories?

L127 – This question is probably related to my lack of knowledge in the earthquake-triggered landslides, but to me it is not clear what you mean here with 'local slope', could you specify? Do you mean the slope at the landslide head ? What is the spatial extent of a "local" slope?

L129 - In Parker et al. 2017, who you cite, they find hillslope gradient as an important driver, which is different than local slope I would think? Parker et al. 2017: "We find that a simple model combining PGA and hillslope gradient provides the most numerically elegant and best fitting model. The use of topographic variables other than hillslope gradient were found to produce models with a lower fit,..."

L135 - Can you add a reference here, after "However, shaking for any future earthquake cannot be predicted due to lack of certainty on source location, magnitude, rupture style, and local site effects.

L194 - How is this "non-local" when accounting for local slope?

L323 - "conditional probability for landslide occurrence" seems more informative to me.

L324 – "Landslide hazard can be defined as..." should already have been clear from the introduction.

L342 - Make reference to preceding research using this approach, yet using rainfall characteristics (I,D) instead of landslide susceptibility (a). E.g., Berti, M., Martina, M. L. V., Franceschini, S., Pignone, S., Simoni, A., & Pizziolo, M. (2012). Probabilistic rainfall thresholds for landslide occurrence using a Bayesian approach. Journal of Geophysical Research: Earth Surface, 117(F4).

L383 - I would strongly reduce this section as readers of NHESS could be assumed to be acquainted with the concept of ROC curves.

L396 – "the naïve (random)" : Necessary to repeat (L394) the two terms here again?

L402 - Why would you use NED elevation data? Since SRTM covers each of the inventory, it seems more logical to use consequently the same DEM source to avoid bias. Certainly because you emphasize on the slope factor here, there should not be a biased introduced voluntarily (unless it would be used for an investigation of sensitivity to spatial resolution)

L416 - Avoid repetition, cf. L181

L420 - Could you clarify what you consider here as channel and channel spacing? How is channel spacing related to the skyline?

L421 - What is meant with 'characteristic hillslope length'?

L423 - What is the relation between the characteristic hillslope length and channel spacing?

L422-423 - Since these are parameterized by the chosen inventories, do you estimate that your rules might change for other areas? Or do you argue that the conservative approach is general enough?

L423 – The sentence "We choose larger window size because skyline angle estimates become asymptotically insensitive to window size" is not clear to me, larger than what?

L437 - Seems to be projected from point P?

L443 - With "non-local" you mean not at the landslide initiation location?

L464 - Avoid repetition with L453.

L547-548 - "on which people generally choose to live" : This statement is too vague to me without a reference, does this statement reflect to your inventories solely?

L567 - I do not see a significant difference in the point density (~number of observations) for observations with Upslope contributing area > $1000 m^2/m$.

L631- Make reference to the respective equations in the Methodology section for the parameters mentioned here.

L673- None, capital N.

L677 - Table 1 and Fig. 6 are redundant, you could add Fig. 6 in supplementary material?

L753-756 – I think it is very valuable that the authors take a step back from there rules while summarizing the main parameters to take into account for hazard assessment, being "hazard area, skyline angle, and the local slope in conjunction with each other". Yet this idea that is stated as a conclusion "We conclude that decisions on how to reduce landslide hazard most effectively need to be made on a case by case basis, …" is not repeated in the abstract or conclusion, which to me is confusing. It is even in contrast with the conclusion stating (L858-859) "suggesting that the average parameters can be applied to other inventories. These findings can be distilled into three simple rules:".

L764-L766 I am not sure what your message is here, helping in decision-making before an earthquake is the same to me as decision making after an earthquake which is in turn also before a future earthquake. What is the differentiation that I am missing here?

L770 - This statement is largely depending on which spatial extent you perform your analysis and therefore I don't think it is relevant, or should be said in a different way.

L849 - In "the highest area at a given slope" it is not clear what you mean with "highest area".

---

## Short Comment (SC2) · 8 Jan 2019

Dear Authors,

I wish to thank you for your reply and for the stimulating discussion. I am looking forward to reading the final version of your manuscript.

Best wishes, Dr. Gianvito Scaringi

---

## Short Comment (SC3) · 8 Jan 2019

**Response to Reviewer 1 (Odin Marc).**
**Many thanks Dr Marc for your constructive and useful review. Your two major comments are both useful we respond to each in bold below. Many of your line by line comments are straightforward so we include responses only where there is more to say.**

MAJOR COMMENTS
My first comment is about the normalization of several of the hazard metrics : I am convinced that a substantial part of the difference between the hazard curves could be removed by plotting the hazard against a landscape metric : For example for slope, each landscape as likely a modal slope, that may be interpreted as the result of geolechanical difference (for steady state landscape at least). Thus curves may be plotted against S-mode(S) , somewhat normalizing for difference between two landscape. I can understand the author may still want to express their rules in terms of absolute values of slope or other variables, but I suspect this normalization would clarify and strengthen the result and their analysis (as this did in other studies). I make suggestion for the other variables in my inline comments.
**This is an excellent suggestion and something that we will certainly explore. As you suggest, we have been keen to express our rules in terms of absolute values to keep them simple and to enable estimates to be made without additional equipment or information. However, your suggested normalization does promise improved explanatory power our only concern is whether reporting this analysis will dilute the focus of the paper.**

My second concern is that their maybe some over-interpretation of the data scatter towards the extremity of the hazard curves. And the author do not provide clear metrics or indication of the validity of individual datapoint. This is not an easy task but the work of Rault et al., which I co-authored, recently proposed a method to do exactly that. I would suggest the author to apply these criterium and check. In this work we consider the probabilty p of the whole topography, and the one resulting from the landslides affected area only p_L. To assess whether p_L is significantly different from p we compute the confidence interval Ip associated to the random drawing of n (n the number of landslides) pixels out of the landscape distribution. If p_L belongs to [p-Ip : p+Ip] then we cannot exclude that the difference between p and p_L just comes from random fluctuations and it is likely not significant. Given landslides remain rare in the whole topography, the drawing can be assumed independent, and similar to a Bernoulli sampling. Provided the central limit theorem is respected (i.e. n>30, np>5 and n(1-p)>5 ) the 90% confidence interval can be estimated as: Ip = p – 1.96 (p(1-p)/n)^0.5 ; p + 1.96 (p(1-p)/n)^0.5.
Some additional details can be found in the supplementary methods of Rault et al., 2018. Basically n is large (n>1000-10,000) so the authors should obtain very narrow Ip until they reach p<0.001 – 0.0001 but I expect these low probability to be reached in the tail of the distribution (Fig 3,4, 6) and the cut off will vary for the different landscape with higher or lower p or n. The authors could compute Ip as well as the convergence criterium and show the points which may be insignificant in shaded / transparent ?
**Thankyou for pointing us to this approach. We had struggled to find a way to account for sample sizes in our analysis and this approach looks perfectly suited to the problem!**

LINE BY LINE COMMENTS
L155: True they pertain to initiation, but vast majority of studies highlighting their role or quantifying statistical relations between these predictors and landslide use total area and therefore are combining both initiation and runout.
**We agree. However, the mechanistic justification for the factors is almost always initiation based. Our point here is that when these variables are used for landslide hazard prediction they are used because to represent controls on landslide initiation. We propose a modification to the text along the following lines: "The potential predictors described above are primarily chosen in hazard models for their perceived link to the probability of coseismic landslide initiation."**

L180: I have the impression it should be the minimum skyline angle, not intersecting topography, …. Indeed a maximum reach angle. Cf comment on Fig 1
**You could phrase this either as: *'The maximum angle from horizontal to the skyline'* or *'the minimum angle from horizontal that does not intersect the skyline'*. We have chosen the former because it is shorter and because we are concerned that the latter is more open to misinterpretation. The misinterpretation would be that people may not think of cones at increasing angles and so may misunderstand or ignore the second clause.**

L458: Shalrun-EQ= Probability of mobilization convolved with connection probability. Average in the above area. So hazard area is basically the number of pixel where debris flow can occur and reach the interest

cell (say Nhaz)... in the contributing area, times pixel area, and divided by the contour length, i.e. the square root of contributing area. Although I am confused because in Fig 2 : Hazard Area seem to be Nhaz times Pixel resolution (or Nhaz.a/sqrt(a) ). But then smallest vales should be 0 and 30 (as it does in Fig 2). But in Fig 6 it goes from 0.1 to 1e3… So there seems to be a problem between the 2 definitions. Please check.

**Your interpretation of SHALRUN-EQ is correct, the reason for hazard areas less than one pixel is our use of multiple flow path routing, which we had not properly explained. We will explain this more clearly.**

L512-516 : Ok, simplicity is important and it is difficult to integrate other effect mentioned here. But what about checking the actual evolution of probability with slope, for both initiation and stop? A reasonable estimate of scar area can be obtained by selecting the highest elevation pixel in your landslide, and selecting as many as needed to reach a scar area with an aspect-ratio of 1.5 (Domej et al., 2017) and a mean width representing of your polygon (see Marc et al., 2018 for how to do that). Doing so you could check if a plateau develop in your probability ratio after 39Æ or so... Interestingly you could reverse the idea and take the lowest N pixel (N ~ Width / 30 for 30m resolution DEM) of your landslides to obtain a probability of stopping.

**We have actually done this analysis already and the results are interesting. They suggest that landslides initiate on a fairly narrow range of slopes but stop on a much broader range (consistent with our observations here) modal values are somewhat similar to the optimized values used here. However, these (unreported) results indicate the slopes on which landslides initiate and stop rather than the probability of initiation and stopping given slope so we need to be careful in connecting the two sets of results. A careful examination of this connection is outside the scope of the paper since we are focused here on coming up with simple rules.**

L536 : Did you check the curve appearance when using gradients, that is tan(Theta) (with theta the slope in degree) ? Because the tan(theta) does appear mostly exponential over a large range of Theta. Thus a linear function of tan(theta) (the relevant parameter for landslide stability) may appear exponential when plotted against theta.

**Yes, the kink in some curves at low slopes suggested that tan(theta) might be a better predictor and it is attractive for its connection to landslide mechanics. However, we found that for most inventories the slope of the tan(theta) relationship was too gentle to provide a good fit to the data at high slopes. Given the additional complexity of the tangent function it is not well suited to a simple rule so we chose not to report it here. However, the misfit between tan(theta) and landslide probability is clearly interesting and merits further examination in future.**

L538-542: Northridge and Haiti are shifted compared to other. They both become > average probability around 20Æ , vs 30 for others. This roughly correspond to modal slopes of these areas. It would be interesting to re-plot all curves not against slope, but slope – Sm the modal slopes. This collapsed curves on a similar analysis for rainfall (cf Marc et al 2018) Similarly is there a large variety of drainage area distribution ? Haiti and Northridge are very peculiar again compared to the other cases. Some normalization by the mode of the landscape drainage area may be important.

**This is a good suggestion in terms of improving the explanation of the dataset as a whole but is problematic in terms of developing a simple rule because it would require knowledge of the slope distribution for a location. We will certainly try the analysis and look for opportunities to report the findings here but we are concerned about retaining our focus on the simple rules. In the case of upslope contributing area we already seek to explain the variability in terms of hillslope length. The best solution may be to point out modal slopes in the same way then include the normalized analysis for both slope and UCA in supplementary information. If not this may need to wait for a paper more focused on the use of these metrics within a more conventional hazard mapping exercise.**

The definition you take for hazard area gives 0 hazard area for the reference in all cases and then a decrease. It does not seem that shift in the horizontal direction would do any good, and the vertical shift seems due to the proportion of zero hazard area in the landscape, so maybe computing a landscape PDF ignoring the zero would be insightful?

**We suspect that normalisation may not be particularly informative in this case, we could introduce a normalisation in each rule i.e. normalized initiation and stopping angles but we are then very far from a simple rule.**

L694: I am a bit surprise by the term of channel inside this rule. I guess it derives from the fact that the hazard consider upslope contributing areas defined from flow algorithm. But the hazard area at many intermediate locations on hillslopes may be a channel for your analysis but not for the resident and deciders of the area. Because a channel is defined on finer scale than the DEM. You already say that this metrics is anyway difficult to estimate and handle for application, but this terminology would also complexify the problem for deciders or policy makers.
**We will think about whether we can come up with a better term, we are talking about areas of convergence so channel seemed to work but perhaps we have to stay general. The key consideration here is that the expression must be precise but also make sense to people on the ground.**

L699 : This is fortunate indeed, almost surprising.
**Agreed we expected more sensitivity to the parameters here.**

L711: Interesting. Do you think this could be somewhat validated by making skyline and hazard graph for landslide above and below a certain threshold (say 5e3 m2 or even better above a certain width...) ?
**The idea of examining whether different metrics preferentially predict larger or smaller landslides is interesting and your suggested approach would be a good way to do that. However, this analysis is probably outside the scope of the current paper given its focus on simple rules. It is difficult to see how our answer to this question could be worked into an improved simple rule.**

L822-23: I do not understand what you mean by "we expect the length scales over which this occurs to be long (order kilometres) relative to the other factors examined here" Do you mean that main lithological units are usually big (regional scales) and thus significant part of a landscape will have homogeneous lithology, whereas topographic attribute change at the scales of 10s of meter ? Then it is the length scale for the variability of lithology that you want to mention. Anyway please clarify.
**Your interpretation is correct but we will try to clarify our point.**

On a side comment, normalizing each landscape slope by their modal slope would be somehow a step toward normalizing difference in landscape that can be due to major lithological or geomechanical attributes (Korup 2008).
**Agreed but as discussed above including this in a simple rule would be problematic.**

L830: And they likely do, given that large landslide (likely to travel further away as you recall in the introduction) are usually reported closer of the fault or at larger shaking values (Khazai and Sitar (2004), for the Chi-Chi earthquake (1999), Massey et al. (2018) for Kaikoura or Valagussa et al 2019 for systematic evaluation of PGA and landslide size distribution. So future exploration of the behavior of your hazard curve split for specific lithology of different area class should be done.
**Agreed, both landslide size and lithology are interesting topics for future work but are outside the scope of this paper.**

L834 : I would say we can reasonably expect strong differences : given that hazard increase strongly with local slope for EQ (Fig 4) but not for the rainfall induced landslides : as shown by the anaysis similar to your Fig 3 in Marc et al., 2018. Further, the longer runout (due to lower stopping angles) and stronger dependence on contributing areas are additional changes.
**We agree that strong differences are possible but we think it is fair to say that the strength of these differences is not yet clear.**

---

## Short Comment (SC4) · 8 Jan 2019

Dear Reviewer,

Thank you for your careful, positive and thorough review of our paper. We are in the process of addressing your comments in full but I wanted to provide an initial response first. We will respond to your general comments here and your in-line comments in the attached file. We have found that a number of your suggestions can be directly implemented and some of the queries simply addressed by slight alterations to wording in the text. As a result, we respond here only where there is more to say.

Many thanks,

David Milledge

[Figure]

GENERAL COMMENT- Thank you for this interesting paper. Using six inventories of coseismic landslides, the authors test the significance of multiple topographical parameters to constrain a set of simple rules in order to minimise exposure to landslide hazard. The paper forms a significant added value to the landslide hazard scientific community as a first attempt in identifying simple rules which is essential for communication about complex hazards to a broad (lay) audience in creating awareness and minimizing landslide exposure. I appreciate the authors' balanced conclusion on the most effective parameters for hazard reduction ["We conclude that decisions on how to reduce landslide hazard most effectively need to be made on a case by case basis, and are best made using hazard area, skyline angle, and the local slope in conjunction with each other."], unfortunately this is not taken in the abstract and conclusion where the authors present without further nuances three simple rules. The discussion is focused on the authors' results with limited reflections with respect to related research (cf. introduction). I believe such a reflection would make the results more convincing. RESPONSE: Thank you for your careful reading of the paper and your many helpful comments and suggestions. We have worked hard to identify this set of simple rules and it is encouraging that this comes through in the manuscript. However, we will take on board your suggestion to temper our presentation of these rules. We will seek to clarify that we are suggesting such rules as a new tool to complement existing approaches rather than replace them, and that these are a first attempt at 'simple rules' with the hope and expectation that others will improve on them in future.

SPECIFIC COMMENTS COMMENT- The first time I read through the paper I found the abstract and introduction confusing while the terms hazard, exposure, risk, hazard response, "anticipating". . . are used without first clearly constraining them. Even though the audience from NHESS should be familiar with these terms I believe that these terms are still easily confused. I would therefore recommend to distinguish these terms in the introduction, or make reference to literature in which this is done. RESPONSE: Thank you for this useful feedback we will seek to clarify this in the revised manuscript.

COMMENT - The paper is well structured and the figures of high quality presenting very clearly the results, yet I would suggest to shorten the paper to bring forward the main messages even more clearly. Sections that I would suggest to reduce are section 4 ("Earthquake inventories") by providing a summary of the used inventories with the most important parameters necessary for the analysis; and section 5 ("Methods") could also be reduced, moreover this would allow the reader to more easily follow the workflow. RESPONSE: We are pleased that you found our presentation of the results clear and will seek to shorten the "Earthquake inventories" section of the manuscript, we see less scope for shortening the methods section but will look for opportunities in that section also.

COMMENT - I wonder how easily the presented rules can be adopted without prior knowledge or skills, which seems to be the main purpose of the study yet lacking from the discussion. This is not easily answered and out of scope of the study to check the applicability of their rules by householders, local government, and NGOs, but I would recommend to be more cautious when claiming to present 'simple rules'. RESPONSE: We have chosen the term 'simple rules' to make the connection to an existing and active field of research around heuristic decision-making. We would strongly argue that the first two rules are simple and do not require prior knowledge or skills: 'minimize your maximum angle to the skyline' and 'avoid steep (>10 degrees) channels with many steep (>39 degrees) areas that are upslope'. Your point here and in detailed comments that the language of the third rule needs to be improved is helpful and we will look for opportunities to do so. Examining the applicability of these rules is, as you suggest, beyond the scope of this study but that doesn't prevent the development and testing of the rules themselves from being a useful exercise.

Please also note the supplement to this comment:
https://www.nat-hazards-earth-syst-sci-discuss.net/nhess-2018-271/nhess-2018-271-SC4-supplement.pdf

[Figure]

[Figure]

**Supplement:**

**Dear Reviewer,**

**Thankyou for your detailed comments. We are in the process of addressing all of them and have found that a number of your suggestions can be directly implemented and some of the queries simply addressed by slight alterations to wording in the text. As a result, we respond here only where there is more to say.**

**Many thanks,**

**David Milledge**

Detailed comments on "Simple rules to minimize exposure to coseismic landslide hazard"

L15 - Do you present in the end primarily simple rules to identify hazard? Or rules to minimize exposure, cf title? I understand they go hand in hand, but it would be good in my opinion to be aware that the terms Hazard, Exposure and Risk are easily confused by readers. Being consequent in using terminology in the abstract might avoid confusion.
**We present rules to identify landslide hazard and we expect their primary use to be in aiding decision making in order to reduce exposure to landslide hazard. We will clarify this in the revised manuscript.**

L20 - From reading only the abstract it is difficult to agree that defining "the upslope area with slope >39° that reaches a location without passing over a slope of <10°" does not require prior knowledge or skills and that it is easy understandable. **Agreed, but on line 23 we distil this into a simpler rule: 'avoid steep (>10˚) channels with many steep (>39˚) areas that are upslope'**

L23 - Show which other metrics were tested besides the two new metrics you introduce so this sentence ("most skilful") has more meaning. **We mention these on lines 16-17 and are conscious that we are short on space in the abstract.**

L26 - How does that work, "minimise local slope especially on steep slopes"? **It is particularly important to minimize local slope on steep slopes. This is explained in more detail in the results and discussion sections. We are not sure if you found the sentence difficult to interpret or were concerned about how robust the finding was. The latter will be addressed in the results and discussion section of the paper.**

L26-28 - This rule seems dubious when stating at the same time "even at the expense of increasing upslope contributing area" and " but not at the expense of [...] hazard area" with the latter also comprising upslope contributing area. **The hazard area is found within the upslope area but these two metrics are very different from one another as we show in the paper. Our results strongly support both parts of the rule that you identify above.**

L46 - I think "respond to that hazard" is of lesser relevance here as you do not deal with hazard response in this paper. **We agree that response is not our focus but information at a scale that enables decisions to be made on how to respond to the hazard is one of the key motivations for this work. Thus, we think it is important to retain the response clause.**

L62 – "hazard maps cannot resolve hazard at those scales" : I doubt that, with the current availability of high-resolution remote sensing data; yet I agree it could be time-consuming. **Agreed, though it is worth highlighting that we are talking about national and regional scale maps in this clause. We will soften the statement by changing from "cannot" to "do not".**

L97 - How does the "self-recovery" relate to the first part of the sentence? I don't see the relevance of it here. **We will add an inline indication of what self-recovery means such as: "self-recovery after disasters (e.g. where householders rebuild their own homes)"**

L102 - Not only of "less use" but also inherently different; your rules aim to minimize landslide exposure, not to help in hazard response. Please modify. **We disagree, action to minimize your exposure to a hazard can occur both before and during an earthquake, taking the earthquake example this might be the**

**difference between relocating away from an earthquake prone area and choosing to 'drop cover and hold on'. Given this we think that 'less use' is the appropriate modifier here.**

L112-116 - I don't see much difference between the two questions? **The first relates to absolute performance of the rule set, the second to relative performance of rules within the set.**

L127 – This question is probably related to my lack of knowledge in the earthquake-triggered landslides, but to me it is not clear what you mean here with 'local slope', could you specify? Do you mean the slope at the landslide head ? What is the spatial extent of a "local" slope?
L129 - In Parker et al. 2017, who you cite, they find hillslope gradient as an important driver, which is different than local slope I would think? Parker et al. 2017: "We find that a simple model combining PGA and hillslope gradient provides the most numerically elegant and best fitting model. The use of topographic variables other than hillslope gradient were found to produce models with a lower fit,..."
**This comment and the comment above are closely related so we will deal with them together. We have used 'local slope' to refer to the same property that Parker et al. 2017 call hillslope gradient. The term local slope is appealing to us since it indicates that a gradient is being calculated over a (relatively) short length scale rather than over the entire hillslope (from ridge to river). However, we will try to clarify this in our revised manuscript.**

L194 - How is this "non-local" when accounting for local slope? **It is non-local in the sense that the value of the metric at a given cell is a function of cells within a wider neighborhood than only its 8 connected (local) neighbors. In this case the property is the gradient (local slope in our terms) of the cells in this wider neighborhood. We see though that local slope is a confusing term here and will seek to clarify this in our revised manuscript.**

L402 - Why would you use NED elevation data? Since SRTM covers each of the inventory, it seems more logical to use consequently the same DEM source to avoid bias. Certainly because you emphasize on the slope factor here, there should not be a biased introduced voluntarily (unless it would be used for an investigation of sensitivity to spatial resolution). **This is a good point, our use of NED data was a legacy of higher resolution analysis at this site. We chose to coarsen the 10 m DEM to 30 m to ensure that the resolution was consistent with other sites. We expect the difference in our results between using NED and SRTM at 30 m to be negligible but we will test this.**

L420 - Could you clarify what you consider here as channel and channel spacing? How is channel spacing related to the skyline?
**We define a channel in geomorphic terms (i.e. as a flowpath with identifiable banks) both because this is recognisible in the field and is the basis for the scaling relationships that we use to define characteristic hillslope length. Channel spacing is related to the window size required to evaluate the skyline angle because the skyline is likely to be defined by local ridges and the distance to these ridges to be defined by channel spacing.**

L421 - What is meant with 'characteristic hillslope length'?
**Characteristic hillslope length can be interpreted as an estimate of the average hillslope length for the study area. It is calculated based on the upslope contributing area at which there is a scaling break in the relationship between slope and upslope contributing area following the approach of Roering et al. (2007).**

L423 - What is the relation between the characteristic hillslope length and channel spacing? **Since channels are separated by ridges with hillslopes on each side then the average channel spacing is twice the characteristic hillslope length.**

L422-423 - Since these are parameterized by the chosen inventories, do you estimate that your rules might change for other areas? Or do you argue that the conservative approach is general enough?
**The size of the window over which the cone is projected should not have an impact on the rules only on their implementation and testing within a GIS. The objective here is to ensure that the chosen window size is large enough to reproduce the same skyline angle in the GIS that would be measured in the field.**

L567 - I do not see a significant difference in the point density (~number of observations) for observations with Upslope contributing area > 1000m./m. **Our point here was that the number of observations per**

**bin was very small for upslope contributing area >1000 m2/m. However, reviewer 1 has suggested an approach that should enable us to identify bins in which sample sizes are too small to confidently interpret.**

L753-756 – I think it is very valuable that the authors take a step back from there rules while summarizing the main parameters to take into account for hazard assessment, being "hazard area, skyline angle, and the local slope in conjunction with each other". Yet this idea that is stated as a conclusion "We conclude that decisions on how to reduce landslide hazard most effectively need to be made on a case by case basis, …" is not repeated in the abstract or conclusion, which to me is confusing. It is even in contrast with the conclusion stating (L858-859) "suggesting that the average parameters can be applied to other inventories. These findings can be distilled into three simple rules:".

**The 'case by case basis' on L754 refers to application of the rules on a case by case rather than simply resolving to always move upslope or downslope for example. As a result we don't see a conflict between these two statements. Hazard should be assessed case by case but this can be done using the simple rules. We will try to clarify this in our revised manuscript.**

L764-L766 I am not sure what your message is here, helping in decision-making before an earthquake is the same to me as decision making after an earthquake which is in turn also before a future earthquake. What is the differentiation that I am missing here?

**The point we are trying to make here is that these rules could be used not only for long-term decision making where the time that it takes to move a certain distance is not the limiting factor in whether you can locate yourself or your assets there but also for short-term decision making during or in the immediate aftermath of an earthquake when you may only be able to move short distances. We will try to clarify this in or revised manuscript.**

---

## Author Comment (AC1) · 8 Jan 2019

Dear Dr Scaringi,

Many thanks for your thoughtful comments and questions, we have tried to respond to each below.

David Milledge

COMMENT- You mentioned multiple times that the DEM resolution can influence some of your results. It would be nice to quantify this influence at least for one inventory for which a higher resolution DEM is available (e.g. Northridge). Perhaps, moving from 30 m to 10 m DEM will only produce marginal improvements while increasing the computational cost significantly, or on the contrary it will change the result significantly.

[Figure]

RESPONSE: This is a good idea and should be straightforward to do. We will attempt it for Northridge, where we have previously tested (but not yet compared results from) the 10 m NED elevation data.

COMMENT - There are cases in which several inventories are available for the same study area (e.g. Wenchuan). These inventories are sometimes quite different from each other. Among others, we discussed this in a recent submission, still under review (see the revised manuscript in the discussion at https://www.earth-syst-sci-data-discuss.net/essd-2018-105/) and we found substantial areal mismatches (up to 67%) between inventories in the Wenchuan, and rather low pixel-based correlations (R-squared as low as 0.35). We showed that this translates in quite some differences in landslide-size probability distributions and hence in landslide volume estimations. This might condition some types of hazard assessments based on volume-runout correlations. However, we did not go deeper into the topic, as it was out of the scope of our manuscript, and we did not investigate how this mismatch between inventories translates into statistics of controlling factors (e.g. slope, upstream contributing area, etc.). It would be interesting if you could estimate to what extent choosing a different inventory for the same study area would affect your assessment.

RESPONSE: We certainly want to follow up on this suggestion and had hoped to report results to you within the discussion period but have been unable to do so. We will attempt it for the revised publication, though in the interests of space, the results may need to go in a supplementary information section unless they have a strong effect on our overall findings.

COMMENT - Also, again about the Wenchuan case, you only chose a subset of the inventory by Li et al. (2014) containing about 1/3 of the landslides. It would be good to explain whether this subset can be thought as representative of the entire study area (e.g. in terms of landslide metrics, topography, lithology, distance from epicentre and fault rupture, etc.) so that one would be confident that the results you obtain have more general validity and are not biased by your choice, which was only due to a data

availability issue. What you report in the conclusion (see my point below), that is that the site-specific and averaged rules perform similarly, is comforting in this sense, but what if it is just a coincidence?

RESPONSE: This is another good suggestion and in the light of it we will make some comparisons to the full study area excluding areas with data gaps. We will also try to give the reader a better sense of the characteristics of the subset relative to those of the full study area.

COMMENT - From your analyses you obtained a set of simple and easily understandable rules to minimise the exposure, and you wrote that the hazard area calculated with averaged parameters performs only slightly worse than hazard area calculated with site-specific parameters. This is encouraging and, as you wrote, it suggests that the average parameters can be applied to other inventories (or subsets of inventories). Thus, it would be very interesting to see these averaged parameters being applied to other inventories, across a variety of landscapes, climates and seismic characteristics. Also, it would be interesting to apply your rules to a highly seismic region in which no recent earthquake has occurred, and relate it to the current distribution of population and exposed goods (but I recognise the latter is out of the scope of this work, so it is just an idea).

RESPONSE: These are both very interesting ideas, though they are out of scope for this work as you say. We are keen to examine these rules in different contexts to establish the range of conditions under which they apply but felt that the six cases used here make a useful initial contribution. We have taken an approach similar to your second idea to provide an indication of the spatial distribution of co-seismic landslides that might be expected in a scenario earthquake for the specific case of an earthquake on the Weinan-Jinyang fault near Xian, China (in prep for IJDRR).

---

## Author Comment (AC2) · 20 Feb 2019

We thank the Dr Scaringi for his interest in our paper and for his useful suggestions, which we feel have considerably improved the article. In our response below his comments are in normal text, and our replies are in bold.

Dear authors,

I enjoyed reading your manuscript, which I believe can be a useful contribution towards landslide risk reduction in highly seismic regions. I have a few questions, mostly regarding the robustness of your findings, which I list as follows:

- You mentioned multiple times that the DEM resolution can influence some of your results. It would be nice to quantify this influence at least for one inventory for which a higher resolution DEM is available (e.g. Northridge). Perhaps, moving from 30 m to 10 m DEM will only produce marginal improvements while increasing the computational cost significantly, or on the contrary it will change the result significantly.

**We have tested the impact of varying DEM resolution from 10 – 90 m for the Northridge study area. We find that performance of slope, skyline angle and upslope contributing area improves slightly at finer resolutions. Hazard area preforms best at the same resolution as that used for parameter optimization (in this case 30 m). Nevertheless, we find that the hazard area metric remains the most skillful predictor of hazard across grid resolutions from 10 m to 60 m, and thus that the rule even when applied over length scales as small as 10 m or as large as 60 m will continue to perform 'well' relative to the alternatives. A description of this test has been added to the Discussion.**

- There are cases in which several inventories are available for the same study area (e.g. Wenchuan). These inventories are sometimes quite different from each other. Among others, we discussed this in a recent submission, still under review (see the revised manuscript in the discussion at https://www.earth-syst-sci-data-discuss.net/essd-2018-105/) and we found substantial areal mismatches (up to 67%) between inventories in the Wenchuan, and rather low pixel-based correlations (R-squared as low as 0.35). We showed that this translates in quite some differences in landslide-size probability distributions and hence in landslide volume estimations. This might condition some types of hazard assessments based on volume-runout correlations. However, we did not go deeper into the topic, as it was out of the scope of our manuscript, and we did not investigate how this mismatch between inventories translates into statistics of controlling factors (e.g. slope, upstream contributing area, etc.). It would be interesting if you could estimate to what extent choosing a different inventory for the same study area would affect your assessment.

**We have tested the impact of different landslide inventories for the Wenchuan earthquake and now report the results in the discussion. We find that the change of inventory has no impact on the rank order of performance of the metrics; and a very minor impact on both the AUC values and the hazard curves. As above, we now provide a description of this test in the Discussion**

- Also, again about the Wenchuan case, you only chose a subset of the inventory by Li et al. (2014) containing about 1/3 of the landslides. It would be good to explain whether this subset can be thought as representative of the entire study area (e.g. in terms of landslide metrics, topography, lithology, distance from epicentre and fault rupture, etc.) so that one would be confident that the results you obtain have more general validity and are not biased by your choice, which was only due to a data availability issue. What you report in the conclusion (see my point below), that is that the site-specific and averaged rules perform similarly, is comforting in this sense, but what if it is just a coincidence?

**Subsetting was necessary because gaps in the SRTM would result in incorrect computations for our topographic metrics, particularly upslope contributing area and hazard area. The subset of landslides that we use run in a swath from north to south. The area extends from the footwall to the hanging wall of the fault crossing the surface expression of the fault and thus spans almost the full range of shaking intensities, lithologies, and topographic settings. Thus, while we cannot rule out the possibility that the site-specific rule for Wenchuan would be different with the full data set, we see no reason why that should be the case. The fact that site-specific and averaged values for hazard area are essentially equivalent also suggests that we are looking at general patterns rather than coincidental relationships. We now include a series of study area maps in the supplementary information showing the study areas and the mapped landslides superimposed on the DEMs. For Wenchuan we show both the full set of landslides mapped by Li et al. (2014) and the subset that we use.**

- From your analyses you obtained a set of simple and easily understandable rules to minimise the exposure, and you wrote that the hazard area calculated with averaged parameters performs only slightly worse than hazard area calculated with site-specific parameters. This is encouraging and, as you wrote, it suggests that the average parameters can be applied to other inventories (or subsets of inventories). Thus, it would be very interesting to see these averaged parameters being applied to other inventories, across a variety of landscapes, climates and seismic characteristics. Also, it would be interesting to apply your rules to a highly seismic region in which no recent earthquake has occurred, and relate it to the current distribution of population and exposed goods (but I recognise the latter is out of the scope of this work, so it is just an idea).

**These are both very interesting ideas, though we feel that they are out of scope for this work as you say. We are keen to examine these rules in different contexts to establish the range of conditions under which they apply, but felt that the six cases used here make a useful initial contribution. We have taken an approach similar to your second idea to provide an indication of the spatial distribution of co-seismic landslides that might be expected in a scenario earthquake for the specific case of an earthquake on the Weinan-Jinyang fault near Xian, China (covered in a separate manuscript submitted to IJDRR).**

---

## Author Comment (AC3) · 20 Feb 2019

We thank the Dr Marc for his careful and helpful review which we feel has considerably improved the article. In our response below reviewer comments are in normal text, and our replies are in bold.

Summary

Milledge et al., present a thorough statistical analysis of six coseismic landslides inventories to relate landslide hazard to landscape properties such as slope and contributing areas, but also more specific variables such as skyline angle and a hazard area integrating the probability of initiation and propagation of landslides. They found that the two latter metrics explain best the location of the inventories and may allow to be converted into simple rules useful for hazard management. The paper is well written, with a straight forward structure and informative. It will make a nice contribution for NHESS both for its systematic analysis and its recommendations. I have two major comments that I think could improve the results and the discussion, and then give a number of minor Line by Line comments with potential clarification or additional small analysis.

Major Comments

My first comment is about the normalization of several of the hazard metrics : I am convinced that a substantial part of the difference between the hazard curves could be removed by plotting the hazard against a landscape metric : For example for slope, each landscape as likely a modal slope, that may be interpreted as the result of geolechanical difference (for steady state landscape at least). Thus curves may be plotted against S-mode(S) , somewhat normalizing for difference between two landscape. I can understand the author may still want to express their rules in terms of absolute values of slope or other variables, but I suspect this normalization would clarify and strengthen the result and their analysis (as this did in other studies). I make suggestion for the other variables in my inline comments.

**This is an excellent suggestion, and indeed we found that normalization collapses the hazard curves to some extent. This is very satisfying in terms of explaining our observations. We include these new results in our revised manuscript though they do not alter our conclusions since normalization does not alter the rank order or improve predictive skill of any metric.**

My second concern is that their maybe some over-interpretation of the data scatter towards the extremity of the hazard curves. And the author do not provide clear metrics or indication of the validity of individual datapoint. This is not an easy task but the work of Rault et al., which I co-authored, recently proposed a method to do exactly that. I would suggest the author to apply these criterium and check.
In this work we consider: the probabilty p of the whole topography, and the one resulting from the landslides affected area only p_L.
To assess whether p_L is significantly different from p we compute the confidence interval Ip associated to the random drawing of n (n the number of landslides) pixels out of the landscape distribution. If p_L belongs to [p-Ip : p+Ip] then we cannot exclude that the difference between p and p_L just comes from random fluctuations and it is likely not significant. Given landslides remain rare in the whole topography, the drawing can be assumed independent, and similar to a Bernoulli sampling. Provided the central limit theorem is respected (i.e. n>30, np>5 and n(1-p)>5 ) the 90% confidence interval can be estimated as:
Ip = p – 1.96 (p(1-p)/n)^0.5 ; p + 1.96 (p(1-p)/n)^0.5. Some additional details can be found in the supplementary methods of Rault et al., 2018. Basically n is large (n>1000-10,000) so the authors should obtain very narrow Ip until they reach p<0.001 – 0.0001 but I expect these low probability to be reached in the tail of the distribution (Fig 3,4, 6) and the cut off will vary for the different landscape with higher or lower p or n. The authors could compute Ip as well as the convergence criterium and show the points which may be insignificant in shaded / transparent ?

**Thank you for pointing us to this approach. We had struggled to find a way to account for sample sizes in our analysis but the Rault et al. approach is extremely well suited to the problem! We have now implemented this method in all cases where we generate hazard curves (i.e. conditional probability curves). In each case (Figs 3-5) we show both those data points that show a significant difference and those that don't, and we explain this distinction in the text.**

Line By line comments:
L123 : I could not find Milledge 2018 in the reference list... please check. **Added.**

L133: Add couple of reference for shaking: e.g. Khazai and Sitar 2004, Meunier 2007. **Added on L141**

L138: I think you should also cite Meunier 2008 here, and probably the recent analysis discussion for an extended number of earthquakes in Rault et al., 2018. **Added on L145.**

L142:152 : A couple of references on the suspected effects would be relevant. Especially the ones cited elsewhere in the text: Parise and Jibson 2000, for lithology, Maufroy et al., 2015 for curvature and ridge amplification. **Added on L154-159.**

L155: True they pertain to initiation, but vast majority of studies highlighting their role or quantifying statistical relations between these predictors and landslide use total area and therefore are combining both initiation and runout.
**We agree. However, the mechanistic justification for the factors is almost always initiation based, as are the GIS approaches that are typically applied to assess landslide susceptibility. Our point here is that when these variables are used for landslide hazard prediction they are used to represent controls on landslide initiation. We have modified the sentence to say:**
**"The potential predictors described above are primarily chosen in hazard models for their perceived link to the probability of coseismic landslide initiation." (L163).**

L180: I have the impression it should be the minimum skyline angle, not intersecting topography, .... Indeed a maximum reach angle. Cf comment on Fig 1
**We could phrase this either as: 'the maximum angle from horizontal to the skyline' or 'the minimum angle from the horizontal that does not intersect the skyline'. We have chosen the former because it is shorter and because we are concerned that the latter is more open to misinterpretation. In particular, people may not think of cones at increasing angles and thus may misunderstand or ignore the second clause.**

L200 – 300: This is certainly at the appreciation of the authors, but I have the impression the earthquake environment (tectonic, climatic, vegetation) is over described. Given you never re-refer to this context later, you may shrink those description and end this section with a sentence like : "these epicentral areas encompasses a large diversity of tectonic (X to Z) , climatic (X to Y) and vegetation cover ( X to Y) contexts, but we assume landslides in all of them should be at first order driven by topographic parameters in the same way".
**Thank you, this is a useful suggestion. We have considerably shortened this section, compressed the information into a table in the Supplementary Information, and added a summary paragraph in line with your suggestion.**

In contrast, some aspects may be missing or insufficiently discussed:
1/ I think the number of landslide polygons used in Chi-Chi is missing. **Agreed, added on L229**
2/ The fact you used a sub-inventories in Wenchuan may mean you artificially limit your analysis to a range of shaking quite different from the other cases . This should be mentionned.
**We agree that this is an important issue and needs clarifying, as also mentioned in our response to Dr Scaringi's comments above. We have added maps of the study areas in the Supplementary Information to help readers interpret our results. In the Wenchuan case we show both the full inventory of Li et al. (2014) and the subset that we use. We have added a statement in the paper itself (L239-242) to show that the range of PGA values experienced in our Wenchuan study area (0.16-1.3 g) is similar to those for the whole inventory (0.12-1.3 g).**

3/ A few words on the Implications of polygon mapping quality on your analysis may be given here (or in discussion), such as the affect of amalgamation ; inclusion or not of debris flow propagation within the river network ? (I know it is a difficult distinction, often ignored but it should impact the statistics, especially of contributing area for example). Implictions of the different resolution limit (for Chi Chi or FInisterre compared to Northridge ) or of the location accuracy ?
**Agreed, we have now added a paragraph on mapping quality in the discussion (L840-853)**

Figure 1: Caption: a cone projected from P no ? **Agreed, modified.**
If your cone as angle define from horizontal upward, are you looking for the minimum skyline angle, not intersecting the topography? That is what I get from your sketch in c). See previous comment about skyline angle definition.
**Both are possible definitions of the angle we are seeking, but we believe our current definition is less open to misinterpretation (see response to comment on line 180 above).**

L372-380: < 10 observations ? Why ? And this is only for the 2 ariable case (Fig 4). For the single variable case it is not clear what is noisy data and where to really set the boundary or unsignificant datapoints. Rault et al., 2018 propose an extension of Meunier et al., 2008 studies with an estimation of the uncertainty of observations based on both the number of observations and the probability. See major comments.
**We have now applied the Rault et al approach, so thank you for pointing it out to us.**

L458: Shalrun-EQ= Probability of mobilization convolved with connection probability. Average in the above area. So hazard area is basically the number of pixel where debris flow can occur and reach the interest cell (say Nhaz)... in the contributing area, times pixel area, and divided by the contour length, i.e. the square root of contributing area. Although I am confused because in Fig 2 : Hazard Area seem to be Nhaz times Pixel resolution (or Nhaz.a/sqrt(a) ). But then smallest vales should be 0 and 30 (as it does in Fig 2). But in Fig 6 it goes from 0.1 to 1e3… So there seems to be a problem between the 2 definitions. Please check.
**Your interpretation of SHALRUN-EQ is correct. The reason for sub-pixel hazard areas is due to multiple flow path routing. We now explain this more clearly (see L432).**

L462: repeat from L452-453. Cut or rephrase ? **Rephrased to avoid repetition.**

L478-482 : Steepest descent may be too conservative, even if your rule needs to be simple, maybe you could mention that probability to propagate on non-steepest descent path is probably non-null.
**This is an error in our explanation. We in fact use multiple flowpath routing, and amended our description of this in the methods section (see L435).**
Also what about landslide large enough to be continuing beyond the first cell with angle below the deposition threshold? I see this is partially acknowledged in the discussion. Maybe you can flag here the fact you discuss such limits later. **We Agree and now point the reader to our discussion of this in section 7.1. (see L450).**

L512-516 : Ok, simplicity is important and it is difficult to integrate other effect mentioned here. But what about checking the actual evolution of probability with slope, for both initiation and stop? A reasonable estimate of scar area can be obtained by selecting the highest elevation pixel in your landslide, and selecting as many as needed to reach a scar area with an aspect-ratio of 1.5 (Domej et al., 2017) and a mean width representing of your polygon (see Marc et al., 2018 for how to do that). Doing so you could check if a plateau develop in your probability ratio after 39Æ or so... Interestingly you could reverse the idea and take the lowest N pixel (N ~ Width / 30 for 30m resolution DEM) of your landslides to obtain a probability of stopping.
**We had performed this analysis with interesting results. We found that landslides initiate on a fairly narrow range of slopes but stop on a much broader range (consistent with our observations), with modal values similar to our optimized values. However, these results are telling us the slopes on which landslides initiate and stop rather than the probability of initiation and stopping given slope, so we need to be careful in connecting the two sets of results. A careful examination of this connection is outside the scope of the paper since we focus on developing simple rules.**

L536 : Did you check the curve appearance when using gradients, that is tan(Theta) (with theta the slope in degree) ? Because the tan(theta) does appear mostly exponential over a large range of Theta. Thus a linear function of tan(theta) (the relevant parameter for landslide stability) may appear exponential when plotted against theta.
**The kink in some curves at low slopes indeed suggested that tan(theta) might be a better predictor. While this is consistent with landslide mechanics, we found that for most inventories this relationship does not provide a good fit to the data at high slopes. Given the additional complexity of the tangent function it is not well suited to a simple rule so we chose not to report it here. However, the misfit between tan(theta) and landslide probability is clearly interesting and merits further examination in the future.**

L538-542: Northridge and Haiti are shifted compared to other. They both become > average probability around 20Æ , vs 30 for others. This roughly correspond to modal slopes of these areas. It would be interesting to re-plot all curves not against slope, but slope – Sm the modal slopes. This collapsed curves on a similar analysis for rainfall (cf Marc et al 2018) Similarly is there a large variety of drainage area distribution ? Haiti and Northridge are very peculiar again compared to the other cases. Some normalization by the mode of the landscape drainage area may be important.

**This is a good suggestion in terms of improving the explanation of the dataset as a whole but does not alter the simple rules because: 1) they are applied in relative terms (i.e. choose the location with the lower local slope); and 2) the alteration would require knowledge of the slope distribution for a location (which will not be available to most users). Nevertheless, we have now performed normalization on both slope and UCA. We include the figures in the supplementary information and briefly report the findings in the main text (L529).**

L542-543: If you consider that Haiti and Northridge are more sensitive because they reach higher ratio it may be a confusion because of the lack of normalization (previous comment). It is plausible the relation between slope – Sm and hazard is similar, only the difference between resolvable slope (with a 30m DEM) and the modal slope is larger, allowing to reach larger relative hazard. I think the effect of normalizing for the landscape must be assessed.
**This is exactly what the normalization shows. We have adjusted the text to reflect this observation, added a normalized panel to the figures, and refer to the normalized results in the text (L505).**

L545: combined or merged PDF rather than amalgamated (that sounds negative an unusual to me but I may be wrong). **Altered to combined (L506).**

L555: You say you observe contributing area, but you have normalized by contour length. In the paragraph about hazard (L489), you say contour length is a^0.5, but it is not so clear what is a (the area of a cell, which cell ?) On Fig 2, contributing area seems to be the square root of a. It would be consistent with the contour length estimated as sqrt(a) but then why not say straight you look at the sqrt of drainage area ? Maybe I missed something, or it is worth clarifying a bit.
**We now clarify this on first introducing upslope contributing area "and normalising by the grid cell width to minimise grid resolution biases" (L371), and in our definition of $l_j$, as the cell width (L461).**

L562: This was somehow my expectation, so why not normalizing the contributing area and thus analyzing a/a_rc , with a_rc the channel ridge transition area? Like this the relative decrease or increase away from this objective characterization of the landscape could be analyzed (and the plot in Fig 3,4 would compare hazard curve shape only, not locations). This seems like an important improvement even if I understand that you may point to the fact a layman user of an hazard rule may not guess the modal slope of its landscape or the value of a_rc. After some analysis in the normalized domain general rules for the natural domain may be derived.
**We agree, and now include a normalized plot showing the Northridge curve partially collapsing onto the other curves. Given the generally poor performance of upslope contributing area we choose not to come up with a new rule based around it, nor adjust the other rules in light of these results.**

Fig 4 is very interesting and make a lot of sense after Fig 3. However, I am wondering about two things…
1/ Would all the plot look the same if you use normalized area and slope ? Maybe not given that it was not expected from Fig 3 that Finisterre would be different, but it seems worth and easy to check.
**The differences that result from normalization are largely in the steepness of the surface rather than the way that slope and upslope contributing area interact. As a result there is little obvious change in Fig 4 as a result of normalization. However, we show the normalized results in the supplementary information for completeness.**

2/ You work with 100 log-bins of a and it seems 1degree bins of slope. So I wonder what is your typical number of DEM cells in each of your bins, and thus how statistically significant bins are… This is a detail as pattern are very consistent and a larger bin size would rapidly increase the amount of data.
**We have applied the approach of Rault et al., extended to 2D, to indicate bins where the hazard is significantly different from the study area average. We flag that in the figure caption.**

Fig 5 : Skyline angle is strongly uni-modal. So I would study all areas with a relative skyline hazard: Sky-Modal(Sky). The modal will account for difference in incision/relief between landscape. A potential outcome of such normalization may be that your case have all similar behavior for high skyline angle (increase and then plateau) but that Gorkha, Haiti, Northridge have a steep decrease below a certain angle while not the three others.
**We tested the effects of normalization and as you suggest it does collapse the data to some extent. We find that normalization is particularly effective at aligning the Gorkha, Haiti and Northridge hazard curves with those from the other sites. We now describe this normalisation in the text.**

The definition you take for hazard area gives 0 hazard area for the reference in all cases and then a decrease. It does not seem that shift in the horizontal direction would do any good, and the vertical shift seems due to the proportion of zero hazard area in the landscape, so maybe computing a landscape PDF ignoring the zero would be insightful?

**We suspect that normalisation may not be particularly informative in this case. We could normalise the initiation and stopping angles but we are then farther from a simple rule and, particularly in the case of stopping angle, it is not entirely clear what the appropriate property to normalize by would be. As a result we do not pursue normalization for hazard area.**

L645: Ah<1 mÇ/m . I am surprised by this threshold, but maybe it is a typo. I would have say in Fig 5b the curves steepens most in all case around 20. It is true that for Haiti, Gorkha and Northridge there is a slight increase in the trend after Ah~1, but minor compare to the later steepening.

**This was a typo, and has now been fixed.**

Also to be sure that the difference between a peak or a plateau is a real result it would be important to check the evolution of the uncertainty in your last bins, where certainly few data are available (even if we cannot read the probability of Ah>1e2 or 1e3).

**A peak followed by a decline in hazard with increasing hazard area is retained within that part of the data where there are sufficient observations to allow confident hazard identification only for Haiti. However, we have adjusted our plots to indicate which observations are more or less certain, using the approach of Rault et al. as described above, and discuss this in the modified text at L630.**

We also do not see the difference in availability of such high hazard area in the different areas, so could a very low availability of such hazard areas in Haiti and Northridge (that have less steep slopes) caused a scattered behavior for Ah>100 instead of Ah>1000. A quantification of uncertainty may clarify that. See major comment. **Your suggestion that the earlier onset of scatter in Northridge and Haiti hazard curves is likely to reflect a lower availability of such steep slopes is supported by the Rault et al. analysis, which clearly identifies the point beyond which the curves become more scattered as the point beyond which hazard cannot be confidently resolved. We make this point in the main text at line 636.**

L672 : This sentence confused me. Do you mean each of the three parameters, may be better than the skyline angle for at least one event ? **Yes, your interpretation is correct. We think the confusion was due to a punctuation error (full stop should have been comma), and we have now fixed this.**

L674: These values do not match Table 1 with 0.72, 0.69, 0.74.... Please correct one or the other. **Thanks for spotting this typo, we have now fixed it.**

L694: I am a bit surprise by the term of channel inside this rule. I guess it derives from the fact that the hazard consider upslope contributing areas defined from flow algorithm. But the hazard area at many intermediate locations on hillslopes may be a channel for your analysis but not for the resident and deciders of the area. Because a channel is defined on finer scale than the DEM. You already say that this metrics is anyway difficult to estimate and handle for application, but this terminology would also complexify the problem for deciders or policy makers.

**We can understand your concern here and have considered alternatives. However, we have chosen to retain the word channel within the rule for two reasons. First, because we feel that it is important to capture the notion of convergence and we are unable to find an alternative wording that can do so. Second, because we expect that if SHALRUN-EQ is calculating convergence using a 30 m DEM, it is extremely unlikely that the real topography does not have some sort of channel or gully within that area. It's hard to imagine a topography that would be convergent at 30 m scale but not obviously channelised or gullied at finer scales.**

L699 : This is fortunate indeed, almost surprising.
**Agreed - we expected more sensitivity to the parameters here.**

L711: Interesting. Do you think this could be somewhat validated by making skyline and hazard graph for landslide above and below a certain threshold (say 5e3 m2 or even better above a certain width...) ?

**This is an interesting idea and something that we will investigate in future but we feel that it is outside the scope of the current paper.**

L739 : You certainly mean Meunier 2008 here. However, note that the new study from Rault et al., 2018 is considerably nuancing these past studies.
**Modified to add citation and account for Rault et al's work (L727).**

L820: And even for a trained observer.
**Agreed but given our simple rules focus we choose to retain the focus on untrained observers here.**

L822-23: I do not understand what you mean by "we expect the length scales over which this occurs to be long (order kilometres) relative to the other factors examined here" Do you mean that main lithological units are usually big (regional scales) and thus significant part of a landscape will have homogeneous lithology, whereas topographic attribute change at the scales of 10s of meter ? Then it is the length scale for the variability of lithology that you want to mention. Anyway please clarify.
**Your interpretation is correct but we have clarified our point in line with your suggestion (see L812 in revised manuscript).**

On a side comment, normalizing each landscape slope by their modal slope would be somehow a step toward normalizing difference in landscape that can be due to major lithological or geomechanical attributes (Korup 2008).
**Agreed, but as discussed above including this in a simple rule would be problematic.**

L824-826: This is an important and natural point to make but I would mention rainfall induced landslides straight here, as area affected by coseismic landslides are often even more often affected by rainfall induced landslides ( at least for wet climate Nepal,Finisterre, Taiwan).
**Agreed, and we have modified this text to: 'such as flooding or even rainfall induced landsliding'. (L815).**

L830: And they likely do, given that large landslide (likely to travel further away as you recall in the introduction) are usually reported closer of the fault or at larger shaking values (Khazai and Sitar (2004), for the Chi-Chi earthquake (1999), Massey et al. (2018) for Kaikoura or Valagussa et al 2019 for systematic evaluation of PGA and landslide size distribution. So future exploration of the behavior of your hazard curve split for specific lithology of different area class should be done.
**Agreed, both landslide size and lithology are interesting topics for future work but are outside the scope of this paper.**

L834 : I would say we can reasonably expect strong differences : given that hazard increase strongly with local slope for EQ (Fig 4) but not for the rainfall induced landslides : as shown by the anaysis similar to your Fig 3 in Marc et al., 2018. Further, the longer runout (due to lower stopping angles) and stronger dependence on contributing areas are additional changes.
**We agree that large differences are possible, but we think it is fair to say that the strength of these differences is not yet clear.**

---

## Author Comment (AC4) · 20 Feb 2019

We thank the reviewer for their careful and helpful review which we feel has considerably improved the article. In our response below reviewer comments are in normal text, and our replies are in bold.

General Comment
Thank you for this interesting paper. Using six inventories of coseismic landslides, the authors test the significance of multiple topographical parameters to constrain a set of simple rules in order to minimise exposure to landslide hazard. The paper forms a significant added value to the landslide hazard scientific community as a first attempt in identifying simple rules which is essential for communication about complex hazards to a broad (lay) audience in creating awareness and minimizing landslide exposure. I appreciate the authors' balanced conclusion on the most effective parameters for hazard reduction ["We conclude that decisions on how to reduce landslide hazard most effectively need to be made on a case by case basis, and are best made using hazard area, skyline angle, and the local slope in conjunction with each other."], unfortunately this is not taken in the abstract and conclusion where the authors present without further nuances three simple rules. The discussion is focused on the authors' results with limited reflections with respect to related research (cf. introduction). I believe such a reflection would make the results more convincing.

**Thank you for your careful reading of the paper and your many helpful comments and suggestions. We have worked hard to identify this set of simple rules and it is encouraging that this comes through in the manuscript. However, we will take on board your suggestion to temper our presentation of these rules. We have sought to clarify that we are suggesting such rules as a new tool to complement existing approaches rather than replace them. We highlight, though, that we are clear from the outset that the rules are designed to complement other approaches. For example, we say in the abstract: "Our simple rules complement, but do not replace, detailed site-specific investigation; they can be used for initial estimation of landslide hazard or guide decision-making in the absence of any other information."**

Specific Comments
The first time I read through the paper I found the abstract and introduction confusing while the terms hazard, exposure, risk, hazard response, "anticipating". . . are used without first clearly constraining them. Even though the audience from NHESS should be familiar with these terms I believe that these terms are still easily confused. I would therefore recommend to distinguish these terms in the introduction, or make reference to literature in which this is done.
**Thank you for this useful feedback. We had been careful to define key terms such as hazard, exposure, risk and mitigation in the introduction and were even concerned that these definitions hampered the flow of the text, so it is helpful to know that they are important. We generally define terms in the introduction rather than the abstract given the limited space in the abstract. In all these cases we give the definition within five lines of introducing the term, though to retain the flow of the text our definitions are generally 'in-line' rather than taking the form of a separate sentence in the form 'x is defined as…'.**
**However, following your comment we have sought to simplify our language, removing hazard response (and instead talking in terms of risk mitigation, which we introduce earlier).**

The paper is well structured and the figures of high quality presenting very clearly the results, yet I would suggest to shorten the paper to bring forward the main messages even more clearly. Sections that I would suggest to reduce are section 4 ("Earthquake inventories") by providing a summary of the used inventories with the most important parameters necessary for the analysis; and section 5 ("Methods") could also be reduced, moreover this would allow the reader to more easily follow the workflow.
**We are pleased that you found our presentation of the results clear. We have considerably shortened the "Earthquake inventories" section of the manuscript and slightly shortened the Methods section.**

I wonder how easily the presented rules can be adopted without prior knowledge or skills, which seems to be the main purpose of the study yet lacking from the discussion. This is not easily answered and out of scope of the study to check the applicability of their rules by householders, local government, and NGOs, but I would recommend to be more cautious when claiming to present 'simple rules'.
**We have chosen the term 'simple rules' to make the connection to an existing and active field of research around heuristic decision-making (e.g. Gigerenzer, 2008). This field explicitly refers to heuristics as 'simple rules' (e.g. Todd and Gigerenzer, 2000, Behavioral and brain sciences,**

**23(5):727-741). We would argue that the first two rules are simple and do not require prior knowledge or skills: 'minimize your maximum angle to the skyline' and 'avoid steep (>10˚) channels with many steep (>40˚) areas that are upslope'.**
**Your point here and in detailed comments that the language of the third rule needs to be improved is helpful and we have simplified this rule to read: 'minimise the angle of the slope under your feet, especially on steep hillsides, but not at the expense of increasing skyline angle or hazard area'.**

**Examining the applicability of these rules is, as you suggest, beyond the scope of this study, but that doesn't prevent the development and testing of the rules themselves from being a useful exercise. We have had some experience of applying these rules with organisations involved in post-earthquake reconstruction in Nepal, and have some positive feedback so far, but it is too early for a more formal evaluation and we feel strongly that this would be the topic of another manuscript.**

Detailed comments on "Simple rules to minimize exposure to coseismic landslide hazard"

L10 - The abstract misses information on the fact that the study is on coseismic landslide hazard.
**Agreed, added 'coseismic' on line 15.**

L15 - Do you present in the end primarily simple rules to identify hazard? Or rules to minimize exposure, cf title? I understand they go hand in hand, but it would be good in my opinion to be aware that the terms Hazard, Exposure and Risk are easily confused by readers. Being consequent in using terminology in the abstract might avoid confusion.
**Thank you for spotting this possible source of confusion. Although the metrics identify hazard, we have written the rules in such a way that they provide advice on action to take to minimize exposure. We have modified this sentence on line 15 the abstract to be consistent with the title.**

L18 - Not sure what you mean with "as a proxy for hillslope location".
**We added: "…location relative to rivers or ridge crests." (L18).**

L20 - From reading only the abstract it is difficult to agree that defining "the upslope area with slope >39° that reaches a location without passing over a slope of <10°" does not require prior knowledge or skills and that it is easy understandable.
**Agreed, but on line 26 we distil this into a simpler rule: 'avoid steep (>10˚) channels with many steep (>40˚) areas that are upslope'**

L22 - Could you add the observation period covered by the inventories here between brackets to know what is 'recent' to you?
**Added: "… earthquakes (occurring between 1993 and 2015)" (L23)**

L23 - Show which other metrics were tested besides the two new metrics you introduce so this sentence ("most skilful") has more meaning.
**We mention these on lines 16-17 and are conscious that we are short on space in the abstract. The text now reads: "We examine rules based on two common metrics of landslide hazard, local slope and upslope contributing area as a proxy for hillslope location relative to rivers or ridge crests. In addition, we introduce and test two new metrics…" (L17)**

L25 - If the rules should be simple and applied by people without skills, why not round to 40°? What is the sensitivity of this rule to a change in the slope of one degree?
**Agreed, we will round to 40 degrees, the impact on performance of a one degree change is negligible.**

L26 - How does that work, "minimise local slope especially on steep slopes"?
**It is particularly important to minimize local slope on steep slopes. This is explained in more detail in the results and discussion sections. We are not sure if you found the sentence difficult to interpret or were concerned about how robust the finding was. The latter will be addressed in the results and discussion section of the paper. We have added a comma between the clauses and altered the second 'slopes' to 'hillsides' to help to clarify the meaning of the phrase (e.g. L28).**

L26-28 - This rule seems dubious when stating at the same time "even at the expense of increasing upslope contributing area" and " but not at the expense of [...] hazard area" with the latter also comprising upslope contributing area.

**The hazard area is found within the upslope area but these two metrics are radically different from one another, as we show in the paper. Our results strongly support both parts of this rule that you identify above.**

L38 - I would suggest to use the updated paper of Petley, 2012:
Froude, Melanie J., and D. Petley. "Global fatal landslide occurrence from 2004 to 2016." Natural Hazards and Earth System Sciences 18 (2018): 2161-2181. Given the very extensive reference list I think that 'e.g. Froude et al. 2018' would do while omitting the other references if not necessary in the rest of the paper.
**Agreed. Added.**

L46 - I think "respond to that hazard" is of lesser relevance here as you do not deal with hazard response in this paper.
**We agree that response is not our focus, but information at a scale that enables decisions to be made on how to respond to the hazard is one of the key motivations for this work. Thus we think it is important to retain the response clause.**

L55 - I would add to "site-specific information that may not be available" something like "such as... " to make it more informative.
**Agreed, changed to "…available (such as geological maps or landslide inventories)". (L56)**

L62 – "hazard maps cannot resolve hazard at those scales" : I doubt that, with the current availability of high-resolution remote sensing data; yet I agree it could be time-consuming.
**Agreed, although it is worth highlighting that we are talking about national and regional scale maps in this clause. We softened the statement by changing from "cannot" to "do not". (L64).**

L97 - How does the "self-recovery" relate to the first part of the sentence? I don't see the relevance of it here.
**We added an inline indication of what self-recovery means "…self-recovery after disasters (for example, via reconstruction programmes in which householders rebuild their own homes)". (L100).**

L102 - Not only of "less use" but also inherently different; your rules aim to minimize landslide exposure, not to help in hazard response. Please modify.
**We disagree with this point. Action to minimize your exposure to a hazard can occur both before and during an earthquake. Taking the earthquake example, this might be the difference between relocating away from an earthquake prone area and choosing to 'drop cover and hold on'. Given this, we think that 'less use' is the appropriate modifier here.**

L107 - Could you site a reference at the end of this sentence, in order to make "our" refer to the scientific background.
**Here, the use of 'our' was referring to the findings of this paper. To clarify this, we modified "Some of our results may be transferrable to landslides caused by more frequent triggers, such as storms, and we consider this point in the discussion." To "We consider the extent to which our results may be transferrable to landslides caused by more frequent triggers, such as storms, in the discussion." (L109).**

L110 - Add respective countries between brackets.
**Modified to: "Finisterre (Papua New Guinnea), Northridge (USA), Chichi (Taiwan), Wenchuan (China), Haiti, and Gorkha (Nepal) earthquakes". (L114).**

L112-116 - I don't see much difference between the two questions?
**The first relates to absolute performance of the rule set, the second to relative performance of rules within the set. We have added this sentence on L121 to clarify this point.**

L116 - What kind of patterns? Temporal/spatial... **modified to "spatial patterns". (L122).**

L118 - Which "combined datasets" you refer to? The landslide inventories or more specifically to the derived topographical parameters from the inventories? **Modified to "landslide datasets". (L124).**

L127 – This question is probably related to my lack of knowledge in the earthquake-triggered landslides, but to me it is not clear what you mean here with 'local slope', could you specify? Do you mean the slope at the landslide head ? What is the spatial extent of a "local" slope?
**We have now clarified our definition of local slope, which although conventional may not be familiar to all readers: "Local slope, the gradient of the ground surface measured over some short distance (usually ~1-100 m)" (L133).**

L129 - In Parker et al. 2017, who you cite, they find hillslope gradient as an important driver, which is different than local slope I would think? Parker et al. 2017: "We find that a simple model combining PGA and hillslope gradient provides the most numerically elegant and best fitting model. The use of topographic variables other than hillslope gradient were found to produce models with a lower fit,..."
**In fact we use 'local slope' to refer to the same property that Parker et al. 2017 call hillslope gradient. We considered a switch to their nomenclature but feel that local slope is the best established and most appropriate term for the property that we refer to. One reason for this is that local slope indicates that a gradient is being calculated over a (relatively) short length scale rather than over the entire hillslope (from ridge to river). It is also more clearly contrasted with a non-local measure like skyline angle, which considers the topography over a larger window around a particular point of interest. We now clarify this by defining local slope within the sentence (on L133) as mentioned above.**

L135 - Can you add a reference here, after "However, shaking for any future earthquake cannot be predicted due to lack of certainty on source location, magnitude, rupture style, and local site effects. Added on L142.**

L194 - How is this "non-local" when accounting for local slope?
**The hazard area is a non-local metric because the value of the metric at a given cell is a function of cells within a wider neighborhood than only its 8 connected (local) neighbors. In this case the property is the gradient (local slope in our terms) of the cells in this wider neighborhood (from all possible initiation points to the target cell).**

L323 - "conditional probability for landslide occurrence" seems more informative to me.
**Agreed, but we are talking about a broader class than simply occurrence. We have modified the title to: "Conditional probability and landslide hazard" (L268)**

L324 – "Landslide hazard can be defined as…" should already have been clear from the introduction.
**Agreed, but here we are building the case for a conditional probability based analysis, so we feel that the connection with the definition of landslide hazard needs to be retained here.**

L342 - Make reference to preceding research using this approach, yet using rainfall characteristics (I,D) instead of landslide susceptibility (a). E.g., Berti, M., Martina, M. L. V., Franceschini, S., Pignone, S., Simoni, A., & Pizziolo, M. (2012). Probabilistic rainfall thresholds for landslide occurrence using a Bayesian approach. Journal of Geophysical Research: Earth Surface, 117(F4).
**Agreed, this is a useful reference and while many other studies apply similar approaches this has a stronger connection than most. We have added: "…landslide inventories. This type of approach has proved successful for a range of applications including identifying topographic controls on vegetation patterns [Milledge et al., 2012] and the rainfall conditions that trigger landslides [Berti et al., 2012]. If we grid…" (L289).**

L383 - I would strongly reduce this section as readers of NHESS could be assumed to be acquainted with the concept of ROC curves. **We feel that a clear explanation of ROC curves is important in this paper because of the central role that these curves play in quantifying the performance of the metrics that we test.**

L396 – "the naïve (random)" : Necessary to repeat (L394) the two terms here again?
**Agreed, removed '(random)' on L356.**

L402 - Why would you use NED elevation data? Since SRTM covers each of the inventory, it seems more logical to use consequently the same DEM source to avoid bias. Certainly because you emphasize on the

slope factor here, there should not be a biased introduced voluntarily (unless it would be used for an investigation of sensitivity to spatial resolution)

**Our approach was to use the best freely-available data at each location, but to use a consistent resolution between sites. For all the locations but Northridge SRTM is the best quality available data. This can be problematic, as SRTM data can have gaps (as in Wenchuan) and can smooth highly dissected terrain (as in Northridge). While in Wenchuan we had to restrict our analysis to a subset of the terrain, in Northridge we were able to use better topographic data (the NED), though we downsampled to the same resolution. Our performance tests at Northridge, comparing SRTM and NED data, support this. We find a considerable performance reduction for SRTM relative to NED data, particularly for the hazard area metric. This is likely due to the highly dissected topography within the Northridge study area; the SRTM data do not capture this topography but the resampled NED data do.**

L416 - Avoid repetition, cf. L181
**Addressed by modifying and shortening sentence.**

L420 - Could you clarify what you consider here as channel and channel spacing? How is channel spacing related to the skyline?
**Channel spacing is related to the window size required to evaluate the skyline angle because the skyline is likely to be defined by local ridges and the distance to these ridges to be defined by channel spacing. However, the term was distracting and in retrospect unnecessary so we have removed it in our new explanation.**

L421 - What is meant with 'characteristic hillslope length'?
**Characteristic hillslope length can be interpreted as an estimate of the average hillslope length for the study area. It is calculated based on the upslope area at which there is a scaling break in the relationship between slope and upslope area following the approach of Roering et al. (2007). We have now replaced 'characteristic' with 'average' since this is a more straightforward term (L382).**

L423 - What is the relation between the characteristic hillslope length and channel spacing?
**Since channels are separated by ridges with hillslopes on each side, then the average channel spacing is twice the characteristic hillslope length. In answering this query we identified an alternative explanation for our choice of search radius that avoids the confusing connection to channels.**

L422-423 - Since these are parameterized by the chosen inventories, do you estimate that your rules might change for other areas? Or do you argue that the conservative approach is general enough?
**The size of this window should not have an impact on the rules. It will affect only on their implementation and testing within a GIS. The objective here is to ensure that the search radius is large enough to reproduce the same horizon angle in the GIS that would be measured in the field.**

**The four comments above suggest that our explanation of our choice of search radius for the skyline angle was a source of confusion. We have now rephrased the entire section as follows (removing reference to channel spacing which was a distraction):**

**"For each cell in a study area, we estimate the skyline angle by calculating vertical angles between the target cell and every other cell within a 4.5 km radius. This search radius is chosen to greatly exceed the average hillslope lengths in all study areas and thus to fully capture the local skyline. The longest average hillslope length out of our study areas is ~500 m for Wenchuan, estimated following the method of Roering et al. (2007). We choose a search radius nine times larger than this hillslope length to ensure redundancy in capturing the local skyline and because the only disadvantage of a larger radius is increased computational cost." L380-386.**

L423 – The sentence "We choose larger window size because skyline angle estimates become asymptotically insensitive to window size" is not clear to me, larger than what?
**This sentence has been removed from the modified manuscript.**

L437 - Seems to be projected from point P?
**Agreed, altered.**

L443 - With "non-local" you mean not at the landslide initiation location?
**This point has been addressed in our earlier discussion of 'non-local'.**

L464 - Avoid repetition with L453.
**Modified to remove repetition.**

L547-548 - "on which people generally choose to live" : This statement is too vague to me without a reference, does this statement reflect to your inventories solely?
**We can be confident of this for our specific inventories but would argue that it is true in general. However, we do not have a reference to support it, so we have adjusted the sentence to refer to our inventories in particular (L509).**

L567 - I do not see a significant difference in the point density (~number of observations) for observations with Upslope contributing area > 1000m./m.
**Our point here was that the number of observations per bin was very small for upslope contributing area >100 m$^2$/m. However, we have adopted a new approach (as suggested by reviewer 1) that enables us to identify the point at which sample sizes per bin are too small to confidently interpret.**

L631- Make reference to the respective equations in the Methodology section for the parameters mentioned here.
**Agreed, equation references added (L615).**

L673- None, capital N.  **The typo was the full stop, which should have been a comma. This is now fixed.**

L677 - Table 1 and Fig. 6 are redundant, you could add Fig. 6 in supplementary material?
**We disagree, and feel that Fig.6 shows the data that are synthesized in Table 1. It is important for readers to see these curves rather than the AUC values only, both because they illustrate the point more clearly than a table of values and because they provide richer information. As a result, we feel that it is important to include this in the text rather than leaving it for the supplementary info.**

L753-756 – I think it is very valuable that the authors take a step back from there rules while summarizing the main parameters to take into account for hazard assessment, being "hazard area, skyline angle, and the local slope in conjunction with each other". Yet this idea that is stated as a conclusion "We conclude that decisions on how to reduce landslide hazard most effectively need to be made on a case by case basis, …" is not repeated in the abstract or conclusion, which to me is confusing. It is even in contrast with the conclusion stating (L858-859) "suggesting that the average parameters can be applied to other inventories. These findings can be distilled into three simple rules:".
**The 'case by case basis' on L754 refers to application of the rules on a case by case rather than simply resolving to always move upslope or downslope for example. This does not conflict with our later conclusions. However, we have modified the sentence on L753 (now L742) to remove the word conclusion and thus avoid confusion.**

L764-L766 I am not sure what your message is here, helping in decision-making before an earthquake is the same to me as decision making after an earthquake which is in turn also before a future earthquake. What is the differentiation that I am missing here?
**The point we are trying to make here is that these rules could be used not only for long-term decision making, where the time that it takes to move a certain distance is not the limiting factor in whether you can locate yourself or your assets, but also for short-term decision making during or in the immediate aftermath of an earthquake when one may only be able to move short distances. We clarify this in our revised manuscript (L752-755).**

L770 - This statement is largely depending on which spatial extent you perform your analysis and therefore I don't think it is relevant, or should be said in a different way.
**Agreed. The sentence order has now been adjusted so that this statement (L759) follows the sentence on the granularity of landslide hazard and is supported by examples in two subsequent sentences.**

L849 - In "the highest area at a given slope" it is not clear what you mean with "highest area".

**Agreed, and this has been rephrased to "largest upslope contributing area" (L735)**.